# Fast Sharpness-escaping Optimization for Long-tailed Learning

## Abstract

Deep neural networks often suffer from poor generalization in long-tailed settings. From a loss landscape perspective, this degradation is largely attributed to the tendency of the optimization process to converge into sharp, unstable minima for underrepresented data. We investigate the recently proposed Muon optimizer, providing theoretical evidence that its gradient orthogonalization balances deterministic update strength across positive-curvature modes, thereby reducing the relative dominance of sharp directions. While effective, the Muon optimizer imposes heavy computational overhead in long-tailed scenarios. To reconcile efficiency with optimization quality in long-tailed learning, we propose Fast Sharpness-escaping Optimization (FaSO). FaSO employs a compositional probabilistic schedule that couples lightweight exploration with increasingly frequent computationally intensive orthogonalized updates. This design targets the escape from sharp minima precisely when it is critical for tail-class generalization. Extensive experiments show that FaSO achieves competitive performance across long-tailed benchmarks while substantially reducing overall computational cost, effectively securing flatter minima and improving generalization for tail classes.

## 1 Introduction

Deep learning has significantly advanced a wide range of domains, from computer vision to large language models, achieving unprecedented performance largely driven by massive, high-quality datasets (Russakovsky et al., 2015). However, real-world data often exhibit long-tailed distributions, where dominant head classes overshadow underrepresented tail classes (Buda et al., 2018). This imbalance causes traditional learning algorithms to bias toward head classes, resulting in poor generalization for tail classes. To address this, various strategies have been proposed, including re-sampling (Chawla et al., 2002), decoupling (Kang et al., 2020), loss re-balancing (Ma et al., 2023; Luo et al., 2024), and contrastive learning (Zhu et al., 2022; Du et al., 2024). While these methods mitigate the dominance of head classes, they often risk overfitting by overexposing limited tail samples, highlighting the need for more robust optimization perspectives.

Recent studies (Rangwani et al., 2022; Li et al., 2025b) provide a crucial geometric perspective, showing that tail classes tend to converge into sharp regions of the loss landscape, characterized by large Hessian eigenvalues. Consequently, these underrepresented classes often exhibit poor generalization performance. A promising direction to address this is Sharpness-Aware Minimization (SAM) (Foret et al., 2021; Liu et al., 2022; Li et al., 2025a), a technique that focuses on escaping sharp minima by minimizing loss in a neighborhood of perturbed weights. However, SAM requires twice the number of backpropagation steps (Zhou et al., 2023a), incurring significant computational costs that hinder its scalability to large-scale datasets and models.

To overcome these challenges without redundant backpropagation, we turn to the recently proposed Muon optimizer (Jordan et al., 2024; Shen et al., 2025), which orthogonalizes gradient updates via Newton-Schulz iteration. We provide theoretical insights demonstrating that Muon's gradient orthogonalization balances update strength across positive-curvature modes, reducing the relative dominance of high-curvature directions that are especially relevant to sharp tail-class landscapes. However, despite its geometric advantages, we observe that Muon introduces non-negligible additional computational overhead in practical training due to

its iterative matrix operations, particularly in scenarios with limited batch sizes or convolutional architectures. This creates a critical trade-off between optimization quality and overall training efficiency.

In this paper, we introduce a novel compositional optimization approach, termed FaSO, to further balance training cost with the benefits of Muon. Prior studies (Fang et al., 2019; Rangwani et al., 2022) and our observations suggest that while SGD effectively navigates early landscapes, it becomes increasingly prone to sharp-region stalling as it behaves more deterministically in later stages. Consequently, FaSO employs a dynamic stochastic schedule that prioritizes efficient SGD during early exploration and progressively transitions to computationally intensive orthogonalized updates as training matures. By concentrating Muon's mode-balancing advantage on the final convergence phase, FaSO mitigates sharpness precisely when it is most critical for the generalization of tail classes, thereby maximizing performance without excessively increasing the computational burden. Our contributions are summarized as follows:

1. We provide loss-landscape insights into Muon, showing that its polar update balances update weights across curvature modes and mitigates the dominance of sharp directions, which is critical for tail-class generalization.

2. We propose a novel method, FaSO, to balance training cost and performance in long-tailed learning. Using a stochastic schedule that progressively increases the probability of orthogonalized updates over training, FaSO helps the model escape sharp regions near convergence without significantly increasing training costs.

3. We conduct extensive experiments across a variety of datasets, demonstrating that both Muon and FaSO generally outperform existing long-tailed learning methods. Notably, FaSO achieves performance comparable to Muon while reducing computational overhead by more than 50%.

## 2 Related Work

### 2.1 Long-tailed Learning

There have been substantial explorations in recent years to address the challenges of long-tailed learning. At the data level, re-sampling (Chawla et al., 2002; He et al., 2008) and data augmentation techniques (Zhang et al., 2018; Yun et al., 2019; Ahn et al., 2023) focus on modifying the training data distribution to mitigate class imbalance. At the representation level, decoupling frameworks (Kang et al., 2020; Xuan & Zhang, 2024) separate the feature learning stage from classifier training, allowing for independent optimization of each component. Multi-expert architectures (Wang et al., 2021b; Tan et al., 2024) employ multiple specialized networks to handle different class groups. Transfer learning approaches (Wang et al., 2021a; Li et al., 2024) enhance the feature space representation for minority classes with knowledge from related domains or tasks. At the loss level, re-weighting techniques (Cui et al., 2019; Luo et al., 2024) assign different weights, while margin-based techniques (Cao et al., 2019; Menon et al., 2021) impose class-specific decision boundaries during training. More recently, contrastive learning frameworks (Cui et al., 2021; Zhu et al., 2022; Du et al., 2024) have demonstrated promising results by encouraging uniformly discriminative feature representations across all classes. However, existing methods often suffer from the risk of overfitting due to limited tail class samples, highlighting the need for more robust optimization methods that can effectively navigate the complex loss landscapes inherent in imbalanced learning scenarios.

### 2.2 Sharpness of Loss Landscape

Generalization remains a central challenge in deep learning, fundamentally determined by the optimization process. Recent studies (Jiang et al., 2020; Stutz et al., 2021; Li et al., 2025b) demonstrate that the loss landscape's geometry is intrinsically linked to generalization, where flatter minima typically generalize better than sharper ones. This relationship is particularly salient in imbalanced learning, where minority class landscapes are frequently dominated by sharp regions (Zhou et al., 2023a). While traditional approaches like Perturbed Gradient Descent (Ge et al., 2015; Jin et al., 2017) employ random noise to escape such regions, they often exhibit suboptimal performance in imbalanced settings due to a lack of sufficient directional guidance (Rangwani et al., 2022). Sharpness-Aware Minimization (SAM) (Foret et al., 2021) offers a more principled alternative by identifying sharp maximal points and explicitly minimizing neighborhood sharpness. This approach has proven effective in promoting flatter convergence for imbalanced data (Rangwani et al.,

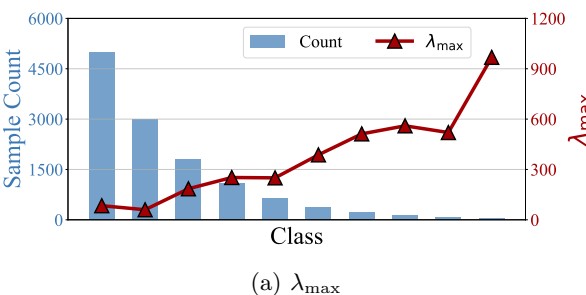
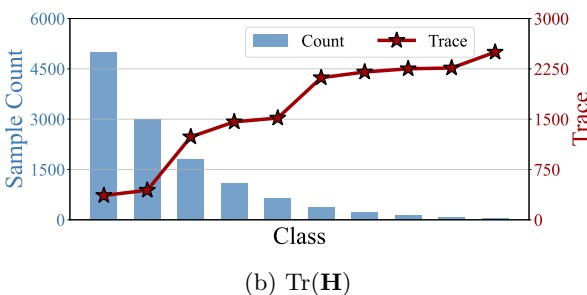

(a) $\lambda_{\max}$                                                                                                          (b) $\mathrm{Tr}(\mathbf{H})$

Figure 1: (a) Maximum eigenvalues $\lambda_{\max}$ ($\downarrow$) and (b) trace of Hessian metric $\mathrm{Tr}(\mathbf{H})$ ($\downarrow$) across classes with different number of samples. Classes with fewer training samples consistently exhibit larger values for both metrics, indicating that these under-represented classes converge to sharper minima in the loss landscape, which can lead to poor generalization performance in long-tailed learning.

2022). Nevertheless, SAM and its variants (Zhou et al., 2023a; Lyu et al., 2025) incur at least double the computational cost of SGD, posing significant scalability bottlenecks where efficiency is paramount.

## 3 Method

In this section, we first establish the preliminaries of our work, including the problem setup and the mechanics of the Muon optimizer. We then present our theoretical analysis of Muon, which serves as the foundation for our work by identifying its capability to balancing positive-curvature mode for sharpness mitigation. Building on these findings, we introduce our primary method, FaSO, a dynamic optimization method designed to leverage these theoretical benefits in a computationally efficient and practically effective manner.

### 3.1 Problem Setup

Let $\mathcal{D} = \{(x_i, y_i)\}_{i=1}^N$ be a training dataset of $N$ samples, where $x_i \in \mathcal{X}$ is an input sample and $y_i \in \mathcal{Y} = \{1, \ldots, C\}$ is its corresponding class label. We denote the number of samples in each class as $\{n_1, \ldots, n_C\}$, and assume, without loss of generality, that $n_i > n_j$ for any $i > j$. Real-world datasets often exhibit a long-tailed distribution with $n_1 \gg n_C$, where a small number of majority classes contain abundant samples while numerous minority classes are data-scarce. Our goal is to learn a deep neural network $h(\cdot; w)$ parameterized by $w \in \mathcal{W}$ that minimizes the empirical risk $\mathcal{L} = \frac{1}{N} \sum_{(x,y) \in \mathcal{D}} \ell(h(x; w), y)$, where $\ell$ denotes a loss function.

### 3.2 Preliminaries

**Review of Muon.** To analyze the optimization dynamics, we focus our discussion, for clarity, on a single parameter matrix $\mathbf{W} \in \mathbb{R}^{m \times n}$. The principles can be extended to the entire parameter set (Kovalev, 2025). We consider two optimization methods: the standard SGD optimizer and the Muon optimizer. For the SGD optimizer, the update rule for a parameter matrix $\mathbf{W}_t$ at iteration $t$ is:

$$\mathbf{W}_{t+1} = \mathbf{W}_t - \eta_t \mathbf{g}_t, \quad \text{where} \quad \mathbf{g}_t = \nabla \mathcal{L}(\mathbf{W}_t). \tag{1}$$

Here, $\eta_t > 0$ is the learning rate and $\mathbf{g}_t$ denotes the stochastic gradient with respect to $\mathbf{W}_t$. For the Muon optimizer, the gradient is first transformed via a Newton-Schulz iteration process and then used to update the parameter $\mathbf{W}_t$. Specifically, the update is performed as:

$$\mathbf{O}_t = \text{Newton–Schulz}(\mathbf{g}_t), \quad \mathbf{W}_{t+1} = \mathbf{W}_t - \eta_t \mathbf{O}_t. \tag{2}$$

The central idea of Muon optimizer is to employ the Newton-Schulz iteration process to approximately compute the polar decomposition $\mathbf{O}_t$ of $\mathbf{g}_t$, which corresponds to $\mathbf{O}_t = \mathbf{U}_t \mathbf{V}_t^T$ in the singular value decomposition (SVD) of $\mathbf{g}_t = \mathbf{U}_t \mathbf{\Sigma}_t \mathbf{V}_t^\top$. Suppose $\mathbf{g}_t \in \mathbb{R}^{m \times n}$ is the gradient matrix with rank $r_t$, $\mathbf{\Sigma}_t \in \mathbb{R}^{r_t \times r_t}$ is a diagonal matrix containing the singular values of $\mathbf{g}_t$, $\mathbf{U}_t \in \mathbb{R}^{m \times r_t}$ and $\mathbf{V}_t \in \mathbb{R}^{n \times r_t}$ are the left and right singular vector

matrices of $\mathbf{g}_t$, respectively. The update matrix becomes $\mathbf{U}_t\mathbf{V}_t^\top$, which represents the closest semi-orthogonal matrix to $\mathbf{g}_t$. Conceptually, this orthogonalization procedure maintains the structural properties of the update matrices, thereby preventing the parameters from being updated along a few dominant directions.

**Newton-Schulz Iteration Process.** This iterative process begins by normalizing the gradient matrix $\mathbf{G}_t = \mathbf{g}_t/\|\mathbf{g}_t\|_\mathrm{F}$, where $\|\cdot\|_\mathrm{F}$ is the Frobenius norm. The iteration is then initialized with $\mathbf{X}_0 = \mathbf{G}_t$, at each step $k$ of the $N$-step iteration, $\mathbf{X}_k$ is updated from $\mathbf{X}_{k-1}$ as:

$$\mathbf{X}_k = a\mathbf{X}_{k-1} + b(\mathbf{X}_{k-1}\mathbf{X}_{k-1}^\mathrm{T})\mathbf{X}_{k-1} + c(\mathbf{X}_{k-1}\mathbf{X}_{k-1}^\mathrm{T})^2\mathbf{X}_{k-1}, \tag{3}$$

where $\mathbf{X}_N$ denotes the final output after $N$ iterative steps. The parameters $a$, $b$, and $c$ are iteration coefficients. To guarantee proper convergence of Equation (3), these coefficients must be tuned such that the polynomial $p(x) = ax + bx^3 + cx^5$ maintains a fixed point in the neighborhood of 1. Following the original formulation (Jordan et al., 2024), we employ the coefficient values $a = 3.4445$, $b = -4.7750$, $c = 2.0315$, and perform 5 iterations. These coefficients are specifically designed to accelerate the convergence rate for matrices with small initial singular values.

### 3.3 Positive-Curvature Mode Balancing for Sharpness Mitigation

**Tail Classes Exhibit Larger Sharpness.** We consider the minimization of a smooth, potentially non-convex objective function $f$ (e.g., cross-entropy loss). The local geometry of the loss landscape is commonly characterized by the spectral properties of the Hessian matrix $\mathbf{H}$. Key indicators of sharpness include the largest eigenvalue $\lambda_\mathrm{max}$ and the trace $\mathrm{Tr}(\mathbf{H})$, where larger value metrics indicate a sharper, more challenging optimization terrain. Following prior work (Rangwani et al., 2022), we empirically investigate this relationship by computing the eigen spectrum of the Hessian for each class on the long-tailed dataset CIFAR-10 LT. As depicted in Figure 1, there is a clear trend where both $\lambda_\mathrm{max}$ and $\mathrm{Tr}(\mathbf{H})$ increase substantially as the number of samples per class decreases. This validates that models trained on tail classes are more prone to converging within sharper regions of the loss landscape. Therefore, an optimizer that mitigates the dominance of high positive-curvature modes is desirable for robust generalization in long-tailed learning.

**Positive-Curvature Mode Balancing of Muon.** In the following, we analyze Muon from the perspective of positive-curvature modes. Following the local curvature view, each mode corresponds to a Hessian eigen-direction, and we use $h_i > 0$ to denote the positive curvature value associated with the $i$-th mode. Recall that Muon replaces the matrix-valued gradient $\mathbf{g}_t = U_t\Sigma_t V_t^\top$ with its polar factor $\mathbf{O}_t = U_t V_t^\top$. This operation removes the singular-value scale of the gradient while preserving its singular directions. Therefore, unlike SGD, whose update strength can be dominated by large-magnitude components, Muon redistributes the update more evenly across spectral modes. In regions where the gradient spectrum is strongly influenced by high positive-curvature directions, this polar transformation reduces the relative dominance of those sharp modes and yields a more balanced update over the positive-curvature spectrum. We formalize this mode-balancing effect in the following proposition, with the detailed proof provided in Appendix B.

**Proposition 3.1.** *Let $\mathbf{g}_t = U_t\Sigma_t V_t^\top$ be the matrix-valued gradient at iteration $t$, and let the Muon update be $\mathbf{O}_t = U_t V_t^\top$. For the $i$-th local positive-curvature mode, let $h_i > 0$ denote its associated positive curvature value. Let $\alpha_i^\mathrm{SGD}$ and $\alpha_i^\mathrm{Muon}$ denote the deterministic update weights assigned to this mode by SGD and the Muon update, respectively. Then, for any two modes $a$ and $b$ with $h_a > h_b > 0$, we have*

$$\frac{\alpha_a^\mathrm{Muon}}{\alpha_b^\mathrm{Muon}} = \sqrt{\frac{h_a}{h_b}} < \frac{h_a}{h_b} = \frac{\alpha_a^\mathrm{SGD}}{\alpha_b^\mathrm{SGD}}. \tag{4}$$

Thus, compared with SGD, Muon compresses the high-to-low positive-curvature weighting ratio. Equivalently, the polar update weakens the relative dominance of high-curvature modes and relatively increases the update weight assigned to lower-curvature modes.

**Remark.** Proposition 3.1 shows that Muon balances matrix-valued updates across positive-curvature modes. In the local analysis, the update weight of SGD along the $i$-th mode scales linearly with the corresponding positive curvature value $h_i$, whereas the update weight of Muon scales as $h_i^{1/2}$. This square-root dependence compresses the gap between sharp and relatively flat directions, making the update less dominated by the

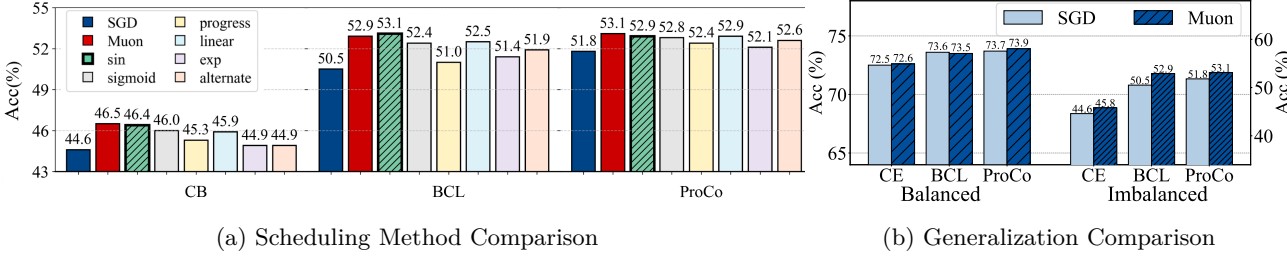

(a) Scheduling Method Comparison

(b) Generalization Comparison

Figure 2: (a) Top-1 accuracy (%) (↑) comparison of various probability scheduling methods. Experiments are conducted on CIFAR-100 LT (IF=100) across various loss functions. Results show that the sinusoidal schedule achieves better performance. (b) Top-1 accuracy (%) (↑) comparison between SGD and Muon on both the balanced CIFAR-100 and CIFAR-100 LT (IF=100) across various loss functions. Results demonstrate that Muon is particularly effective for improving generalization performance in long-tailed settings.

sharpest modes while still preserving progress along the positive-curvature spectrum. Since $\lambda_{\max}$ measures the strongest positive-curvature mode and $\text{Tr}(\mathbf{H})$ summarizes the aggregate positive-curvature mass, this mode-balancing effect is aligned with the empirical sharpness metrics in Figure 1. This provides a geometric explanation for why Muon can guide optimization toward flatter regions in long-tailed learning.

### 3.4 FaSO: A Compositional Optimization Approach

**Computational Overhead Analysis.** While Proposition 3.1 explains how Muon balances updates across curvature modes to mitigate sharpness, its computational overhead presents practical challenges in long-tailed learning. For further analysis, we estimate the FLOP overhead introduced by the Newton-Schulz iteration. For a linear layer parameterized by a weight matrix $\mathbf{W} \in \mathbb{R}^{m \times n}$, each Newton-Schulz iteration requires approximately $6mn^2$ FLOPs. For $T$ iterations, this amounts to $6Tmn^2$ FLOPs. The standard linear layer computation involves approximately $6mnL$ FLOPs, where $L$ represents the number of inputs processed (Jordan et al., 2024). For linear layers $L = B$, where $B$ denotes the batch size in tokens. Thus, the FLOP overhead $\Delta\mathcal{F}_{\text{linear}}$ for a linear layer can be estimated as:

$$\Delta\mathcal{F}_{\text{linear}} = \frac{T \cdot 6mn^2}{L \cdot 6mn} = \frac{Tn}{B} \in \mathcal{O}\left(\frac{Tn}{B}\right). \tag{5}$$

For a convolutional layer, the kernel is flattened into an $m \times n$ matrix for optimization, where $m = C_{\text{out}}$ is the number of output channels and $n = C_{\text{in}} \cdot k^2$ is the product of the input channels and kernel size. The number of inputs per step is $L = B \cdot H_{\text{out}} \cdot W_{\text{out}}$, where $H_{\text{out}}$ and $W_{\text{out}}$ are the spatial dimensions (height and width) of the output tensor. The FLOP overhead $\Delta\mathcal{F}_{\text{conv}}$ is then:

$$\Delta\mathcal{F}_{\text{conv}} = \frac{T \cdot 6mn^2}{L \cdot 6mn} = \frac{TC_{\text{in}}k^2}{BH_{\text{out}}W_{\text{out}}} \in \mathcal{O}\left(\frac{TC_{\text{in}}k^2}{BH_{\text{out}}W_{\text{out}}}\right). \tag{6}$$

Prior studies (Jordan et al., 2024) have shown that Muon maintains FLOPs overhead below 1% in large-scale language model training, where token counts per batch can reach millions (e.g., 16M tokens in LLaVA-405B). However, long-tailed recognition tasks typically employ much smaller batch sizes. This discrepancy introduces a new challenge: while Muon proves effective at mitigating sharpness, its computational overhead can become non-negligible in certain scenarios. For instance, in a ResNet layer with $C_{\text{in}} = 512, k = 3$, and an output feature map of $7 \times 7$, using $T = 5$ iterations with a batch size of $B = 256$ would result in an estimated overhead of 183% according to Equation (6). This substantial increase in training time could limit the scalability of using Muon.

**Training Dynamics Insights.** To balance computational efficiency with optimization performance, we focus on the training dynamics of SGD. Prior research (Fang et al., 2019; Rangwani et al., 2022; Abbe et al., 2023) indicates that in the early stages of training, the inherent stochasticity of SGD provides sufficient noise

to effectively navigate away from sharp areas, but it behaves increasingly like deterministic gradient descent as training progresses and the learning rate decays, making it more prone to stalling near sharp regions late in training, especially in long-tailed scenarios. This observation suggests that the capability of Muon to reach flatter minima is most valuable during later training phases when SGD's inherent noise becomes insufficient. For more empirical observations, please refer to Appendix C.8.

**Compositional Optimization Approach.** Building on this insight, we propose FaSO, a compositional framework that probabilistically schedules between SGD and Muon updates based on training progress. Specifically, given a total training duration of $T$ epochs, we define a selection probability $p_t \in [0, 1]$ for the current epoch $t$ via a scheduling function $h : [0, 1] \rightarrow [0, 1]$, such that $p_t = h(t/T)$, where $h(\cdot)$ is a continuous, non-decreasing function. At each training iteration, the optimizer stochastically determines the update rule: it applies the Muon update with probability $p_t$, and the SGD update with probability $1 - p_t$. We formalize this by defining a composite gradient update operator $\tilde{\mathbf{g}}_t$. The expected update direction and the resulting parameter update rule are given by:

$$\mathbb{E}[\tilde{\mathbf{g}}_t] = (1 - p_t)\mathbf{g}_t + p_t\mathbf{O}_t, \tag{7}$$

$$\mathbf{W}_{t+1} = \mathbf{W}_t - \eta_t\tilde{\mathbf{g}}_t. \tag{8}$$

This monotonically increasing probability schedule ensures that SGD is predominantly selected during the early training phase (*i.e.,* Eq. (1)). This approach leverages SGD's inherent stochasticity for efficient exploration while mitigating the computational overhead associated with Muon. As training progresses, the probability of selecting Muon gradually increases (*i.e.,* Eq. (2)), providing the necessary geometric support to mitigate sharp positive-curvature dominance precisely when SGD's noise becomes insufficient to prevent stalling. This dynamic approach maximizes the generalization benefits of Muon while maintaining high computational efficiency. Notably, both optimizers operate on the same parameter set, with Muon simply applying stateless orthogonalization to the gradients without introducing additional state or parameters.

**Sinusoidal as Schedule Function.** To further assess the appropriate design for $h(\cdot)$, we investigated several candidates: (1) *Linear*, where the probability increases linearly from 0 to 1; (2) *Phased*, which employs SGD exclusively in the first half of training and switches entirely to Muon in the second half; (3) *Exponential*, where the probability grows exponentially from 0 to 1; (4) *Sigmoid*, where the probability grows following a sigmoid function from 0 to 1; (5) *Sinusoidal*, where the probability grows following the first quarter-period of a sine wave; (6) *Alternating*, which alternates between Muon and SGD at each epoch. The results on CIFAR-100 LT with an imbalance factor of 100 are presented in Figure 2a. While these dynamic schedules generally outperform SGD, we observe that the sinusoidal schedule consistently achieves superior performance and stability across various loss functions. This finding highlights that gradually biasing the training process toward Muon in a sinusoidal manner offers more stability and adaptability. Consequently, we adopt the sinusoidal schedule as the default schedule function, i.e., $h = \sin(\pi t/2T)$. The pseudo-code for the training processes of Muon and our FaSO under this schedule are provided in Appendix A.

## 4 Experiments

### 4.1 Experimental Setup

**Datasets.** We evaluate the proposed Muon optimizer method on a suite of widely used long-tailed benchmarks: CIFAR-10 LT, CIFAR-100 LT, ImageNet-LT, and Places-LT. CIFAR-10 LT and CIFAR-100 LT are two long-tailed datasets sampled from the original CIFAR datasets (Krizhevsky et al., 2009). We conduct experiments under varying imbalance factors, defined as IF $= n_{\max}/n_{\min}$. Following the mainstream protocol (Cui et al., 2019), we adopt imbalance settings with imbalance factors of 10 and 100 for both datasets, where the number of samples per class decreases exponentially. Large-scale evaluations are conducted on the ImageNet-LT dataset (115.8k images across 1,000 categories) (Liu et al., 2019) derived from the ImageNet dataset (Russakovsky et al., 2015) and Places-LT (62.5k images across 365 categories) (Liu et al., 2019).

**Evaluation Protocol.** Following standard protocols in long-tailed learning (Wang et al., 2023), we report top-1 accuracy across *Many*, *Medium*, and *Few* splits (Cui et al., 2019; Rangwani et al., 2022). To assess computational efficiency, we also record the average training time per epoch associated with each method.

Table 1: Top-1 accuracy (%) (↑) results on *Many*, *Medium* (namely Med.), *Few*, and overall classes on CIFAR-10 LT and CIFAR-100 LT datasets across various loss functions, categorized by imbalance factors (IF) of 10 and 100. FaSO and Muon are highlighted in light gray for focused comparison.

| Loss | Method | CIFAR-10 LT IF=100 | | | | CIFAR-100 LT IF=10 | | | | CIFAR-100 LT IF=100 | | | |
|---|---|---|---|---|---|---|---|---|---|---|---|---|---|
| | | Many | Med. | Few | All | Many | Med. | Few | All | Many | Med. | Few | All |
| CE | SGD | 94.1 | 77.4 | 65.0 | 77.4 | 75.6 | 62.8 | 48.2 | 60.8 | 75.9 | 52.0 | 15.7 | 44.6 |
| | SAM | 95.7 | 76.7 | 64.4 | 77.5 | 76.7 | 64.4 | 49.0 | 61.9 | 77.5 | 51.1 | 15.8 | 44.9 |
| | **FaSO** | 95.2 | 76.9 | 64.3 | 77.3 | 77.1 | 65.4 | 49.7 | **62.6** | 77.2 | 53.9 | 16.2 | **45.8** |
| | **Muon** | 95.1 | 75.7 | 67.1 | **78.1** | 75.0 | 65.6 | 49.1 | 61.8 | 77.2 | 52.4 | 17.3 | **45.8** |
| CB | SGD | 94.8 | 77.0 | 66.0 | 77.9 | 75.1 | 63.5 | 48.4 | 60.9 | 75.0 | 50.6 | 17.3 | 44.6 |
| | SAM | 94.9 | 76.0 | 65.8 | 77.6 | 77.5 | 65.1 | 48.3 | 62.1 | 75.4 | 50.6 | 19.0 | 45.4 |
| | **FaSO** | 94.9 | 77.1 | 66.7 | **78.3** | 76.1 | 66.4 | 50.5 | **62.9** | 76.5 | 52.5 | 19.3 | 46.4 |
| | **Muon** | 95.1 | 77.9 | 66.0 | **78.3** | 76.4 | 65.9 | 50.6 | **62.9** | 76.4 | 52.2 | 19.7 | **46.5** |
| LA | SGD | 90.3 | 76.9 | 80.9 | 82.5 | 70.0 | 64.3 | 57.1 | 63.2 | 69.2 | 53.6 | 34.3 | 50.5 |
| | SAM | 91.9 | 78.2 | 81.9 | 83.8 | 72.6 | 64.5 | 58.8 | 64.6 | 67.6 | 54.6 | 35.8 | 51.0 |
| | **FaSO** | 91.5 | 78.0 | 82.0 | 83.7 | 71.9 | 65.0 | 58.6 | 64.5 | 69.3 | 55.0 | 34.8 | 51.2 |
| | **Muon** | 92.6 | 79.9 | 82.4 | **84.7** | 71.7 | 65.0 | 59.3 | **64.7** | 68.5 | 56.2 | 36.1 | **51.9** |
| BCL | SGD | 93.2 | 79.3 | 81.7 | 84.4 | 71.7 | 64.5 | 59.5 | 64.7 | 68.5 | 54.2 | 34.2 | 50.5 |
| | SAM | 94.0 | 80.8 | 82.7 | 85.5 | 72.5 | 65.2 | 60.0 | 65.3 | 68.1 | 53.5 | 37.1 | 51.3 |
| | **FaSO** | 93.9 | 80.2 | 82.1 | 85.1 | 73.9 | 66.0 | 60.4 | **66.1** | 71.1 | 57.5 | 36.3 | **53.1** |
| | **Muon** | 94.3 | 80.5 | 82.9 | **85.6** | 73.2 | 66.3 | 60.5 | 66.0 | 70.7 | 56.4 | 36.9 | 52.9 |
| ProCo | SGD | 93.6 | 80.7 | 82.2 | 85.2 | 71.8 | 64.7 | 59.2 | 64.6 | 68.9 | 55.4 | 36.2 | 51.8 |
| | SAM | 92.6 | 80.3 | 84.7 | 85.8 | 73.6 | 64.2 | 59.9 | 65.3 | 69.4 | 56.2 | 36.7 | 52.4 |
| | **FaSO** | 94.2 | 81.0 | 83.3 | **85.9** | 73.6 | 67.3 | 59.6 | 66.1 | 70.1 | 57.1 | 36.9 | 52.9 |
| | **Muon** | 94.2 | 81.6 | 82.9 | **85.9** | 73.8 | 65.4 | 61.5 | **66.4** | 70.0 | 57.4 | 37.2 | **53.1** |

**Baselines.** We compare our method with a range of strong baselines commonly used in long-tailed classification. We evaluate four optimizers: SGD, SAM, Muon, and our proposed FaSO, applied to various widely used methods: Cross-Entropy (CE), Class-Balanced Loss (CB) (Cui et al., 2019), Logit Adjustment (LA) (Menon et al., 2021), Balanced Contrastive Learning (BCL) (Zhu et al., 2022), and Probabilistic Contrastive Learning (ProCo) (Du et al., 2024). This allows us to comprehensively assess the contribution of our optimizer across various long-tailed learning paradigms.

**Implementation details.** Our code is implemented with PyTorch 1.12.1. All experiments are carried out on NVIDIA GeForce RTX 3090 GPUs. For a fair comparison, we use ResNet32 on CIFAR-10 LT and CIFAR-100 LT, ResNet50 on ImageNet-LT, and pre-trained ResNet-152 on Places-LT. We train each model using a batch size of 256 (for CIFAR-10 LT and CIFAR-100 LT) / 128 (for ImageNet-LT) / 512 (for Places-LT), with a momentum of 0.9 and a weight decay of 0.0002. We adopt the Nesterov momentum form for all optimizers, with an initial learning rate of 0.1; a multi-step schedule (decayed to 0.01 and 0.0001 at epochs 160 and 180) for CIFAR-10 LT and CIFAR-100 LT, and a cosine schedule throughout training for ImageNet-LT and Places-LT. For Newton-Schulz iteration steps $N$ in the Muon optimizer, we set $N = 5$ for efficiency.

### 4.2 Comparison Results

**Results on CIFAR-10 LT and CIFAR-100 LT.** We first evaluate Muon and FaSO on CIFAR-10 LT and CIFAR-100 LT under imbalance factors (IF) of 10 and 100. As shown in Table 1, both methods generally improve over SGD and achieve competitive or better performance than SAM, with the clearest gains on tail classes under severe imbalance. On CIFAR-10 LT, Muon improves overall accuracy across baselines and often boosts tail accuracy, particularly at IF=100. The advantage is even more pronounced on CIFAR-100 LT: Muon not only improves overall accuracy under both moderate and extreme imbalance, but also delivers substantial gains for tail classes; for instance, when paired with the CB loss, Muon improves tail class accuracy by 2.2% (IF=10) and 2.4% (IF=100) over SGD. Compared with SAM, Muon is especially strong under the

Table 2: Top-1 accuracy (%) (↑) results on *Many*, *Medium* (namely Med.), *Few*, and overall classes on ImageNet-LT (namely IN-LT) and Places-LT (namely PL-LT) datasets, categorized by different loss functions (CE, LA, and ProCo). FaSO and Muon are highlighted in light gray for focused comparison.

| Dataset | Method | CE | | | | LA | | | | ProCo | | | |
|---|---|---|---|---|---|---|---|---|---|---|---|---|---|
| | | Many | Med. | Few | All | Many | Med. | Few | All | Many | Med. | Few | All |
| IN-LT | SGD | 69.4 | 42.2 | 14.8 | 49.0 | 64.3 | 52.4 | 35.1 | 54.6 | 66.3 | 54.3 | 37.8 | 56.7 |
| | SAM | 71.7 | 43.7 | 16.1 | 50.7 | 66.1 | 54.5 | 38.5 | 56.8 | 66.8 | 56.9 | 40.2 | 58.5 |
| | **FaSO** | 72.7 | 45.1 | 16.2 | **51.8** | 67.4 | 54.2 | 37.8 | 57.1 | 68.4 | 56.6 | 41.1 | **59.0** |
| | **Muon** | 72.5 | 44.1 | 16.1 | 51.2 | 68.5 | 54.5 | 37.4 | **57.6** | 67.3 | 56.0 | 39.5 | 58.1 |
| PL-LT | SGD | 46.3 | 22.0 | 4.4 | 27.3 | 42.0 | 40.3 | 27.4 | 38.4 | 43.6 | 42.0 | 26.4 | 39.5 |
| | SAM | 47.0 | 25.2 | 9.1 | 29.9 | 42.1 | 42.2 | 33.3 | 40.4 | 42.9 | 42.6 | 30.3 | 40.3 |
| | **FaSO** | 47.0 | 25.2 | 9.1 | 29.9 | 43.3 | 41.7 | 32.5 | **40.5** | 43.4 | 42.1 | 33.0 | **40.8** |
| | **Muon** | 47.6 | 26.9 | 10.7 | **31.2** | 43.4 | 41.6 | 33.1 | **40.5** | 43.4 | 42.2 | 31.9 | 40.6 |

Table 3: Loss landscape geometry metrics for SGD and Muon on CIFAR-10 LT and CIFAR-100 LT with imbalance factor 100. Maximum eigenvalue $\lambda_{\max}$ (↓) and trace of Hessian matrix $\mathrm{Tr}(\mathbf{H})$ (↓) are reported for the three least frequent classes. Superscripts (1), (2), and (3) denote the 1st, 2nd, and 3rd rarest classes, respectively. Lower values indicate flatter minima and improved generalization.

| Dataset | Method | $\lambda_{\max}^{(1)}$ | $\lambda_{\max}^{(2)}$ | $\lambda_{\max}^{(3)}$ | $\mathrm{Tr}(\mathbf{H})^{(1)}$ | $\mathrm{Tr}(\mathbf{H})^{(2)}$ | $\mathrm{Tr}(\mathbf{H})^{(3)}$ |
|---|---|---|---|---|---|---|---|
| CIFAR-10 LT | SGD | 968.51 | 516.88 | 560.69 | 2499.73 | 2263.09 | 2251.20 |
| | **Muon** | **116.02** | **115.23** | **95.00** | **404.58** | **377.77** | **412.38** |
| CIFAR-100 LT | SGD | 929.56 | 524.80 | 404.30 | 1560.40 | 1604.30 | 1843.20 |
| | **Muon** | **358.20** | **309.71** | **322.10** | **437.58** | **529.98** | **810.12** |

more severe IF=100 setting and remains competitive under IF=10, indicating robust performance across various settings.

Crucially, our proposed FaSO not only matches but sometimes even surpasses the performance of the Muon optimizer across various experimental settings. This indicates that our compositional optimization method successfully captures the benefits of Muon's gradient orthogonalization during critical training phases while maintaining computational efficiency (Table 4). Additionally, as shown in Figure 2, experiments on both the balanced and imbalanced versions of CIFAR-100 demonstrate that Muon is particularly effective in enhancing generalization performance under imbalanced settings. See Appendix C for more comparison results.

**Results on Large-Scale Datasets.** The benefits of Muon become more pronounced on large-scale datasets, which present far more extreme class imbalance and substantially more classes. As detailed in Table 2, Muon delivers consistent and notable gains, particularly for the underrepresented medium and tail classes. On Places-LT, Muon significantly improves overall accuracy over SGD by 1.1% to 3.9% across various loss functions. Critically, its impact is most profound on the tail classes, boosting their accuracy by up to 6.3% (with CE loss). Furthermore, Muon consistently achieves superior or competitive performance compared to SAM, demonstrating its ability to find superior generalizing solutions. These trends hold on ImageNet-LT. Muon again surpasses SGD, with overall accuracy improving by 1.4% to 3.0%. The benefit for tail classes remains significant, confirming the robustness of our method under diverse, challenging conditions.

Notably, our proposed FaSO continues to exhibit strong performance on large-scale benchmarks across diverse data regimes, showing a favorable trend over both SGD and SAM. Its accuracy is largely on par with the Muon optimizer. Intriguingly, when combined with the most effective loss function, ProCo, FaSO achieves competitive or even superior performance compared to the Muon optimizer on both datasets. This suggests that the dynamic scheduling of optimizers may introduce a more diverse optimization pathway, potentially guiding the model towards wider, better-generalizing minima than either optimizer could find alone.

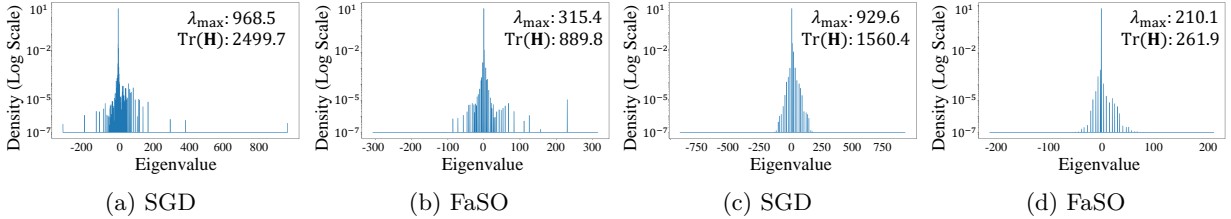

(a) SGD      (b) FaSO      (c) SGD      (d) FaSO

Figure 3: Eigen spectral density for the class with the fewest samples across different methods. Experiments are conducted on (a,b) CIFAR-10 LT and (c,d) CIFAR-100 LT with an imbalance factor 100. Maximum eigenvalue $\lambda_{\max}$ ($\downarrow$) and trace of Hessian metric $\mathrm{Tr}(\mathbf{H})$ ($\downarrow$) in the top right corner of each panel. Lower $\lambda_{\max}$ and $\mathrm{Tr}(\mathbf{H})$ indicate a smoother loss landscape and improved generalization.

Table 4: Computational overhead of different optimizers (SGD, SAM, Muon, and FaSO) on long-tailed benchmarks including CIFAR-100 LT, ImageNet-LT, and Places-LT. We report the average training time per epoch (seconds) ($\downarrow$) and the runtime ratio relative to SGD. FaSO and Muon are highlighted in light gray to group them for focused comparison against the baselines.

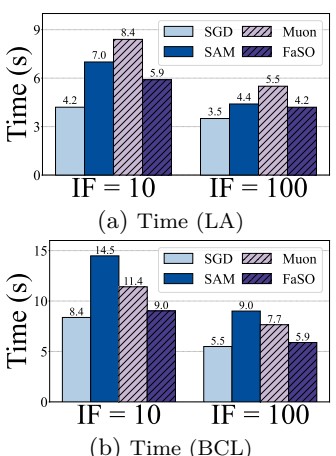

(a) Time (LA)

(b) Time (BCL)

Figure 4: Average training time per epoch (s) ($\downarrow$) for various methods on CIFAR-100 LT with IF of 10 and 100, using (a) LA and (b) BCL.

| Loss | Method | CIFAR-100 LT | | ImageNet-LT | Places-LT |
|------|--------|------|------|-------------|-----------|
| | | IF=10 | IF=100 | | |
| LA | SGD | 4.2s (1.00×) | 3.5s (1.00×) | 184.8s (1.00×) | 174.0s (1.00×) |
| | SAM | 7.0s (1.67×) | 4.4s (1.27×) | 392.2s (2.21×) | 216.0s (1.24×) |
| | **Muon** | 8.4s (1.99×) | 5.5s (1.61×) | 358.8s (1.94×) | 472.8s (2.71×) |
| | **FaSO** | 5.9s (1.41×) | 4.2s (1.23×) | 244.2s (1.32×) | 204.0s (1.17×) |
| ProCo | SGD | 11.1s (1.00×) | 7.0s (1.00×) | 684.0s (1.00×) | 622.8s (1.00×) |
| | SAM | 21.3s (1.92×) | 13.0s (1.85×) | 1870.2s (2.73×) | 2829.0s (4.54×) |
| | **Muon** | 17.6s (1.59×) | 11.3s (1.60×) | 1118.4s (1.63×) | 1608.0s (2.58×) |
| | **FaSO** | 12.4s (1.12×) | 8.2s (1.17×) | 804.0s (1.17×) | 874.8s (1.40×) |

**Flatness of Loss Landscape.** To further investigate the mechanism behind the improved generalization performance of tail classes, we examine the optimization through the lens of the loss landscape. We compute the eigenvalue spectrum of the Hessian matrix for tail classes on both CIFAR-10 LT and CIFAR-100 LT datasets with an imbalance factor of 100, training with the LA loss, as shown in Table 3 and Figure 3. Table 3 presents the Hessian properties for the three classes with the smallest sample sizes, comparing SGD and Muon optimizers through key metrics such as the maximum eigenvalue $\lambda_{\max}$ and trace $\mathrm{Tr}(\mathbf{H})$ at convergence. Smaller values indicate flatter loss landscapes associated with better generalization. The results show clear advantages of Muon: on CIFAR-10 LT, $\lambda_{\max}$ drops by 77%–88% and the trace by 81–84% relative to SGD. On the more challenging CIFAR-100 LT, reductions remain substantial, with decreases of 20%–61% in $\lambda_{\max}$ and 56%–72% in the trace. These findings indicate that Muon effectively drives tail classes toward flatter minima, which is consistent with our theoretical analysis.

**Computational Efficiency Analysis.** We conduct a comprehensive analysis of the computational efficiency of FaSO against the SGD, SAM, and Muon optimizers, as detailed in Table 4 and Figure 4. In Table 4, we measure the average training time per epoch across four representative datasets and the runtime ratio relative to SGD, thereby providing a more fine-grained view of efficiency, using two representative loss functions, LA and ProCo, to evaluate the robustness of each optimizer under varying loss complexities.

The results demonstrate that FaSO effectively resolves the trade-off between generalization and computational cost, achieving the strong performance of Muon with minimal overhead. This efficiency is most striking with complex losses such as ProCo, where FaSO adds only 21% training overhead compared to SGD. In sharp

contrast, SAM incurs a 176% overhead, and Muon still bears a considerable 85%, indicating pronounced scalability limitations when integrated with advanced long-tailed learning methods. When paired with LA loss, FaSO maintains its advantage, incurring just 25% overhead versus 36% for SAM and 106% for Muon. Crucially, these efficiency gains come at no cost to accuracy. As shown in Tables 1 and 2, FaSO generally matches the performance of the Muon optimizer, establishing it as a practical and scalable method for real-world, large-scale long-tail learning. Please refer to Appendix C.2 for additional comparison results.

Table 5: Ablation study on the number of Newton-Schulz iteration steps ($N$) on CIFAR-100 LT under an imbalance factor of 100. We report top-1 accuracy (%) ($\uparrow$) on *Many*, *Medium* (namely Med.), *Few*, and overall classes, along with average training time per epoch (seconds) ($\downarrow$) and the runtime ratio relative to SGD.

| Method | $N$ | Many | Med. | Few | All | Time/epoch |
|---|---|---|---|---|---|---|
| SGD | - | 75.0 | 50.6 | 17.3 | 44.6 | 2.57 (1.00×) |
| **Muon** | 5 | 76.4 | 53.1 | 19.7 | **46.7** | 4.11 (1.59×) |
| **FaSO** | 5 | 76.5 | 52.5 | 19.3 | 46.4 | 2.94 (1.14×) |
| FaSO | 4 | 77.2 | 51.3 | 18.8 | 46.1 | 2.82 (1.10×) |
| FaSO | 3 | 76.1 | 52.8 | 18.1 | 45.9 | 2.78 (1.09×) |
| FaSO | 2 | 76.1 | 51.1 | 17.7 | 45.2 | 2.59 (1.01×) |

**Efficiency Analysis via Gradient Approximation.** To further mitigate computational overhead, we investigate reducing the precision of Newton-Schulz orthogonalization as a potential optimization for efficient gradient approximation. Specifically, we evaluate FaSO on CIFAR-100 LT by varying the number of iteration steps $N$ from 5 to 2. As summarized in Table 5, the results reveal a clear trade-off between computational budget and predictive performance; consistent with our theoretical complexity analysis (Eq. (5,6)), reducing iterations successfully lowers the computational overhead. While the default $N = 5$ yields the highest accuracy, performance consistently remains superior to SGD even as $N$ declines, validating that moderate gradient approximation is an effective method for maintaining robust performance in resource-constrained scenarios.

## 5 Conclusion

In this work, we present a geometric analysis of the Muon optimizer, showing that it balances updates across curvature directions to mitigate sharp tail-class landscapes. To harness this for long-tailed learning, we introduce FaSO, which employs a compositional probabilistic schedule that couples lightweight exploration with increasingly frequent orthogonalized updates as training approaches convergence. This design effectively balances computational efficiency with generalization, guiding models toward flatter landscapes for under-represented classes, as validated by extensive experiments. Future work will extend our approach to other imbalanced domains, such as domain adaptation, to further enhance its applicability and robustness.

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

## A  Appendix

## A  Algorithm

We present the pseudo-code of Muon and FaSO in Algorithm 1 and Algorithm 2, respectively, to illustrate the detailed implementation procedure of our method.

---

**Algorithm 1** Muon
___

**Input**: Initial weights $\mathbf{W}_0$, learning rate schedule $\{\eta_t\}$, momentum $\beta$, batch size $B$, dataset $\mathcal{D}$
**for** $t = 0$ to $T_{\max} - 1$ **do**
    Sample mini batch $\{\xi_{t,i}\}_{i=1}^B \leftarrow \mathcal{D}$
    Calculate $\mathbf{g}_t = \frac{1}{B} \sum_{i=1}^B \nabla f(\mathbf{W}_t; \xi_{t,i})$
    If $t > 0$, $\mathbf{M}_t = \beta \mathbf{M}_{t-1} + (1 - \beta)\mathbf{g}_t$. If $t = 0$, $\mathbf{M}_0 = \mathbf{g}_0$
    Calculate $\mathbf{O}_t = \text{NewtonSchulz}(\mathbf{M}_t)$
    Update $\mathbf{W}_{t+1} = \mathbf{W}_t - \eta_t \mathbf{O}_t$
**end for**

---

**Algorithm 2** FaSO
___

**Input**: Initial weights $\mathbf{W}_0$, learning rate schedule $\{\eta_t\}$, momentum $\beta$, batch size $B$, dataset $\mathcal{D}$
**for** $t = 0$ to $T_{\max} - 1$ **do**
    Sample mini batch $\{\xi_{t,i}\}_{i=1}^B \leftarrow \mathcal{D}$
    Calculate $\mathbf{g}_t = \frac{1}{B} \sum_{i=1}^B \nabla f(\mathbf{W}_t; \xi_{t,i})$
    If $t > 0$, $\mathbf{M}_t = \beta \mathbf{M}_{t-1} + (1 - \beta)\mathbf{g}_t$. If $t = 0$, $\mathbf{M}_0 = \mathbf{g}_0$
    Calculate $p_t$ via sinusoidal schedule
    Sample $\mu \sim \text{Uniform}(0, 1)$
    If $\mu < p_t$ then $\mathbf{O}_t = \text{NewtonSchulz}(\mathbf{M}_t)$ else $\mathbf{O}_t = \mathbf{M}_t$
    Update $\mathbf{W}_{t+1} = \mathbf{W}_t - \eta_t \mathbf{O}_t$
**end for**

---

## B  Theoretical Supplement

### B.1  Proof of Proposition 3.1

We focus on one matrix parameter $\mathbf{W}_t \in \mathbb{R}^{m \times n}$ and denote its matrix-valued gradient by $\mathbf{g}_t = \nabla f(\mathbf{W}_t)$. Let $\mathbf{g}_t = U_t \Sigma_t V_t^\top$ be its rank-$r$ SVD, where $U_t \in \mathbb{R}^{m \times r}$, $V_t \in \mathbb{R}^{n \times r}$, and $\Sigma_t \in \mathbb{R}^{r \times r}$ contains the positive singular values. The Muon update replaces $\mathbf{g}_t$ with its polar factor $\mathbf{O}_t = U_t V_t^\top$. We first rewrite this update in an equivalent preconditioned form. Since $\mathbf{g}_t \mathbf{g}_t^\top = U_t \Sigma_t^2 U_t^\top$ is positive semidefinite, its Moore–Penrose inverse square root is $\left((\mathbf{g}_t \mathbf{g}_t^\top)^\dagger\right)^{1/2} = U_t \Sigma_t^{-1} U_t^\top$. Therefore,

$$\left((\mathbf{g}_t \mathbf{g}_t^\top)^\dagger\right)^{1/2} \mathbf{g}_t = U_t \Sigma_t^{-1} U_t^\top U_t \Sigma_t V_t^\top \\ = U_t V_t^\top = \mathbf{O}_t, \tag{9}$$

where we used $U_t^\top U_t = \mathbf{I}_r$.

We next compare how SGD and Muon assign deterministic update strength across local positive-curvature modes. Let $\mathbf{W}^\star$ be a nearby local reference point and define the residual matrix as $\mathbf{R}_t = \mathbf{W}_t - \mathbf{W}^\star$. Around $\mathbf{W}^\star$, a local quadratic expansion approximates the gradient by the Hessian action on the residual. Following the one-sided Kronecker-factored approximation used in K-FAC-style local curvature models (Martens & Grosse, 2015), we use

$$\mathbf{g}_t = \nabla f(\mathbf{W}_t) \approx \theta_t \mathbf{H} \mathbf{R}_t, \tag{10}$$

where $\theta_t > 0$ is a scalar factor and $\mathbf{H}$ is an effective positive-curvature matrix acting on the left-side modes of $\mathbf{W}_t$. On the considered positive-curvature subspace, we write

$$\mathbf{H} = \mathbf{Q}\operatorname{diag}(h_1, \ldots, h_m)\mathbf{Q}^\top, \qquad h_i > 0. \tag{11}$$

Here, the columns of $\mathbf{Q}$ define the local positive-curvature modes, and $h_i$ denotes the positive curvature value associated with the $i$-th mode. The scalar $\theta_t$ captures the global curvature scale, while the relative magnitudes of $h$ characterize how curvature is distributed across these modes.

We first consider SGD. By Equation (10), the deterministic SGD update direction is proportional to $\mathbf{H}\mathbf{R}_t$. Let $\bar{\mathbf{R}}_t = \mathbf{Q}^\top \mathbf{R}_t$. Using Equation (11), we have

$$\begin{aligned}
\mathbf{Q}^\top \mathbf{H} \mathbf{R}_t &= \mathbf{Q}^\top \mathbf{Q} \operatorname{diag}(h_1, \ldots, h_m) \mathbf{Q}^\top \mathbf{R}_t \\
&= \operatorname{diag}(h_1, \ldots, h_m) \bar{\mathbf{R}}_t.
\end{aligned} \tag{12}$$

For the mode-wise comparison, let $\alpha_i^{\text{SGD}}$ and $\alpha_i^{\text{Muon}}$ denote the coefficients multiplying the same residual component in the $i$-th mode after removing global scalar factors shared across modes. Then Equation (12) shows that the deterministic update weight of SGD along the $i$-th mode is $\alpha_i^{\text{SGD}} = h_i$.

We next consider Muon. Following the common use of gradient second moments and Fisher-type matrices as practical curvature proxies (Kunstner et al., 2019; Martens, 2020; Wu et al., 2026), we use the local relation $\mathbf{g}_t \mathbf{g}_t^\top \approx \omega_t \mathbf{H}$, where $\omega_t > 0$ is a scalar factor. Combining this relation with Equation (9) gives

$$\begin{aligned}
\mathbf{O}_t &\approx \omega_t^{-1/2} \mathbf{H}^{-1/2} \mathbf{g}_t \\
&\approx \omega_t^{-1/2} \mathbf{H}^{-1/2} \left(\theta_t \mathbf{H} \mathbf{R}_t\right) \\
&= \theta_t \omega_t^{-1/2} \mathbf{H}^{1/2} \mathbf{R}_t.
\end{aligned} \tag{13}$$

The scalar factor $\theta_t \omega_t^{-1/2}$ only changes the global update scale and can be absorbed into the effective learning rate. Since $\mathbf{H}^{1/2} = \mathbf{Q}\operatorname{diag}(h_1^{1/2}, \ldots, h_m^{1/2})\mathbf{Q}^\top$, we obtain

$$\mathbf{Q}^\top \mathbf{H}^{1/2} \mathbf{R}_t = \operatorname{diag}\left(h_1^{1/2}, \ldots, h_m^{1/2}\right) \bar{\mathbf{R}}_t. \tag{14}$$

Consequently, after removing the same residual component and global scalar factors, the deterministic update weight of Muon along the $i$-th mode is $\alpha_i^{\text{Muon}} = h_i^{1/2}$.

Now consider two positive-curvature modes $a$ and $b$ with $h_a > h_b > 0$. Since $\alpha_i^{\text{SGD}} = h_i$ and $\alpha_i^{\text{Muon}} = h_i^{1/2}$, we have

$$\frac{\alpha_a^{\text{Muon}}}{\alpha_b^{\text{Muon}}} = \sqrt{\frac{h_a}{h_b}} < \frac{h_a}{h_b} = \frac{\alpha_a^{\text{SGD}}}{\alpha_b^{\text{SGD}}}, \tag{15}$$

where the inequality follows from $h_a/h_b > 1$. The result shows that, under the local positive-curvature comparison above, the Muon update compresses the high-to-low positive-curvature update-weight ratio compared with SGD. Therefore, the polar update reduces the relative dominance of high-curvature modes and produces a more balanced deterministic update over the considered positive-curvature spectrum.

To complement the local curvature approximations used above, Table 6 provides additional empirical support on CIFAR-10 LT and CIFAR-100 LT, measuring the stability of the local curvature structure in Equation (10) and its alignment with the gradient second-moment proxy used in Equation (13).

## C   Experimental Supplement

### C.1   Additional Experiments on CIFAR

Table 7 presents the comparative performance of Muon and FaSO on the CIFAR-10 LT dataset with an imbalance factor of 10. The results demonstrate that both Muon and FaSO consistently surpass the SGD and SAM baselines across various loss functions, aligning with the trend observed in Table 1.

Table 6: **Empirical support for the local curvature approximations.** We evaluate the classifier weights on CIFAR-10 LT and CIFAR-100 LT under IF=100. Stability is measured by comparing the curvature-mode ordering and normalized curvature drift between epochs 180 and 200; high Spearman values and low drift indicate that the local curvature structure remains stable within this training window. Alignment is measured by comparing the leading-mode ordering between the curvature proxy and the gradient second-moment proxy; high Spearman values indicate that the gradient second-moment proxy follows the same curvature-mode structure used in the Muon update analysis.

| Dataset | Stability Spearman | Stability Drift (%) | Alignment Spearman |
|---|---|---|---|
| CIFAR-10 LT | $0.997 \pm 0.008$ | $8.4 \pm 1.1$ | $0.988 \pm 0.013$ |
| CIFAR-100 LT | $0.991 \pm 0.005$ | $5.0 \pm 0.9$ | $0.953 \pm 0.017$ |

Table 7: Top-1 accuracy (%) ($\uparrow$) results of different optimizers under various loss functions on CIFAR-10 LT with an imbalance factor of 10. Results for the *Medium* class group are presented as Med. in the table. FaSO and Muon are highlighted in light gray for focused comparison.

| Loss | Method | Many | Med. | Few | All |
|---|---|---|---|---|---|
| CE | SGD | 95.0 | 85.9 | 88.2 | 89.3 |
| | SAM | 95.2 | 86.3 | 88.1 | 89.1 |
| | **FaSO** | 95.0 | 86.4 | 89.9 | **90.0** |
| | **Muon** | 96.1 | 86.3 | 88.3 | 89.8 |
| CB | SGD | 94.9 | 86.4 | 88.4 | 89.6 |
| | SAM | 95.0 | 86.0 | 87.9 | 89.3 |
| | **FaSO** | 95.7 | 86.6 | 88.6 | 89.9 |
| | **Muon** | 95.6 | 87.2 | 88.7 | **90.2** |
| LA | SGD | 93.8 | 87.5 | 92.1 | 90.8 |
| | SAM | 94.1 | 87.0 | 92.1 | 90.7 |
| | **FaSO** | 94.5 | 87.5 | 92.2 | 91.0 |
| | **Muon** | 94.5 | 88.0 | 92.6 | **91.3** |
| BCL | SGD | 94.3 | 87.5 | 91.8 | 90.8 |
| | SAM | 94.5 | 88.3 | 93.2 | 91.6 |
| | **FaSO** | 95.0 | 88.6 | 92.5 | **91.7** |
| | **Muon** | 94.8 | 88.2 | 92.2 | 91.4 |
| ProCo | SGD | 94.8 | 88.6 | 92.6 | 91.7 |
| | SAM | 94.6 | 88.9 | 93.1 | 91.8 |
| | **FaSO** | 95.3 | 88.7 | 92.7 | 91.9 |
| | **Muon** | 94.8 | 88.6 | 93.5 | **92.0** |

Table 8: Computational overhead of different optimizers under CE loss on long-tailed benchmarks. We report the average training time per epoch (seconds) ($\downarrow$) and the runtime ratio relative to SGD (in parentheses). FaSO and Muon are highlighted in light gray for focused comparison.

| Method | CIFAR-100 | | ImageNet-LT | Places-LT |
|---|---|---|---|---|
| | IF=10 | IF=100 | | |
| SGD | 3.47s (1.00×) | 3.10s (1.00×) | 280.046s (1.00×) | 224.70s (1.00×) |
| SAM | 6.94s (1.99×) | 4.66s (1.50×) | 392.170s (1.40×) | 356.08s (1.58×) |
| **Muon** | 8.22s (2.37×) | 5.87s (1.89×) | 435.29s (1.55×) | 471.63s (2.10×) |
| **FaSO** | 4.49s (1.29×) | 3.35s (1.08×) | 292.51s (1.04×) | 267.29s (1.19×) |

## C.2 Additional Experiments on Computational Overhead

In Table 8, we provide additional experiments analyzing computational efficiency. We measure the average training time per epoch across four datasets using the CE loss function. The results show that the SAM optimizer incurs an average of 98% additional training time compared to SGD, while the Muon optimizer increases training time by an average of 106% under the same settings. In contrast, our proposed FaSO increases training time by only 15% on average relative to SGD. These findings are consistent with the results presented in Table 4 and Figures 3a and 3b.

## C.3 Generalization to Natural Language Processing

While our primary evaluation followed mainstream long-tailed learning protocols centered on visual benchmarks (Menon et al., 2021), we further investigated the versatility of Muon by extending our experiments to the Natural Language Processing domain. We conducted experiments using the Yahoo Answers Topic Classification dataset (Zhang et al., 2015). To simulate long-tailed distributions, we constructed two variants by sampling from a 12k training subset with imbalance factors of 10 and 50, respectively. We divide the classes into *Many*, *Medium*, and *Few* splits, corresponding to the top three, middle four, and bottom three classes sorted by frequency, respectively. Evaluation was performed on a balanced test set containing 4k samples. The model architecture consisted of a fixed pre-trained BERT-base-uncased backbone, followed by an MLP layer and a linear classification head. Both SGD and Muon were trained using CE loss for 20 epochs. As shown in Table 9, Muon consistently outperforms the SGD baseline across different imbalance factors, especially in the tail classes and highly imbalanced setting. This confirms that the benefits of Muon's curvature-aware optimization are not limited to vision tasks but also extend effectively to other modalities like NLP.

Table 9: Top-1 accuracy (%) (↑) results for *Many*, *Medium*, *Few*, and overall classes on long-tailed Yahoo Answers dataset, categorized by imbalance factors (IF) of 10 and 50.

| IF | Method | Many | Medium | Few | All |
|---|---|---|---|---|---|
| 10 | SGD | 75.9 | 58.5 | 43.3 | 59.1 |
| | **Muon** | 73.0 | 60.9 | 45.6 | **59.9** |
| 50 | SGD | 74.8 | 61.2 | 3.4 | 47.9 |
| | **Muon** | 76.8 | 55.9 | 17.3 | **50.6** |

## C.4 Comparison with Fine-Tuning Method

Recent long-tailed recognition methods have explored fine-tuning paradigms on top of large-scale foundation models, such as LIFT (Shi et al., 2024) and LPT (Dong et al., 2023). To verify that Muon remains effective in this setting, we follow the experimental protocol of LIFT. Specifically, we adopt a pre-trained CLIP ViT-B/16 backbone and fine-tune it on CIFAR-100 LT with IF=100. We adhere to the experimental settings of LIFT for a fair comparison. As shown in Table 10, Muon achieves higher overall accuracy than LIFT, with particularly notable gains on tail classes. This indicates that Muon is complementary to fine-tuning based long-tailed methods, and can further improve representation quality even when starting from strong pre-trained features.

Table 10: Top-1 accuracy (%) (↑) results for *Many*, *Medium*, *Few*, and overall classes on CIFAR-100 LT under IF=100 with a CLIP ViT-B/16 backbone.

| Method | Many | Medium | Few | All |
|---|---|---|---|---|
| LIFT | 84.4 | 81.1 | 74.4 | 80.2 |
| **Muon** | 85.1 | 81.5 | 76.8 | **81.3** |

### C.5  Muon with Decoupled Training Method

We further explore the performance of Muon when combined with decoupled training methods. Specifically, we evaluate Muon and FaSO under the standard two-stage decoupling framework (Kang et al., 2020): (1) *Stage 1*, trains the backbone representation, and (2) *Stage 2*, re-trains a balanced classifier (cRT) on top of the frozen backbone. Concretely, we train the backbone in Stage 1 using SGD, Muon, or FaSO, and then apply classifier re-training (cRT) in Stage 2. Experiments are conducted on CIFAR-100 LT under IF=10 and IF=100.

As shown in Table 11, Muon and FaSO consistently outperform SGD after cRT, indicating that re-balancing the classifier does not diminish their advantages. Instead, the gains persist because Muon and FaSO improve the quality of learned representations during Stage 1, providing a stronger feature space for the balanced classifier in the later stage.

Table 11: Top-1 accuracy (%) (↑) results for *Many*, *Medium*, *Few*, and overall classes on CIFAR-100 LT, categorized by imbalance factors (IF) of 10 and 100. FaSO and Muon are highlighted in light gray for focused comparison.

| IF | Method | Many | Medium | Few | All |
|----|--------|------|--------|-----|-----|
| 10 | SGD | 67.3 | 61.5 | 55.9 | 61.8 |
| | **Muon** | 68.3 | 61.9 | 57.3 | 62.8 |
| | **FaSO** | 68.9 | 62.4 | 56.5 | **62.9** |
| 100 | SGD | 66.3 | 51.7 | 31.7 | 47.5 |
| | **Muon** | 66.9 | 56.3 | 34.0 | **50.6** |
| | **FaSO** | 66.1 | 54.8 | 34.3 | 50.0 |

### C.6  Comparison with Strong SAM Variant

**Comparison with ImbSAM.** Several SAM variants have been proposed for long-tailed learning, such as ImbSAM (Zhou et al., 2023a). We further conduct additional comparisons to evaluate the effectiveness and efficiency of our proposed FaSO. We compare FaSO against SGD, SAM, and ImbSAM on CIFAR-100-LT under imbalance factors IF=10 and IF=100. Following prior work, we consider both CE loss and BCL loss. We also report the training time measured on a single NVIDIA RTX 3090, normalized by the SGD baseline to highlight the efficiency trade-off.

As shown in Table 12, FaSO consistently attains the best overall accuracy across all settings while being substantially more efficient than SAM and ImbSAM. In particular, ImbSAM requires roughly 2.0×-3.1× the training cost of SGD due to its extra gradient computations, whereas FaSO only incurs a marginal overhead of about 1.1×–1.3×. These results demonstrate that FaSO achieves a more favorable accuracy-efficiency trade-off than computationally heavy SAM variants in long-tailed recognition.

**Comparison with LookSAM.** We further compare Muon and FaSO with LookSAM (Liu et al., 2022), a representative efficient SAM variant in balanced scenarios. We compare these methods on CIFAR-100-LT under an imbalance factor of 100. The results are shown in Table 13. Although we verified that LookSAM matches SAM's performance on the balanced CIFAR-100 (both achieve an accuracy of 72.8%), the results show that its accuracy deteriorates substantially on the imbalanced CIFAR-100 LT, particularly on tail classes. In contrast, FaSO retains the robust generalization. This indicates that efficiency techniques effective in balanced settings, such as the gradient decomposition and estimation strategies used in LookSAM, are not robust under severe class imbalance. These findings underscore the value of developing efficient alternatives that remain effective in imbalanced scenarios, such as Muon and FaSO.

### C.7  Results on Large-Scale Real-World Long-Tailed Dataset

To provide a more comprehensive evaluation, we extend our experiments to the large-scale real-world setting. We additionally benchmark our method on the iNaturalist-2018 (Horn et al., 2018) dataset. The

Table 12: Top-1 accuracy (%) (↑) results for *Many*, *Medium*, *Few*, and overall classes on CIFAR-100 LT with CE and BCL losses, under imbalance factors (IF) of 10 and 100. We also report the training time (seconds) (↓) and the runtime ratio relative to SGD (in parentheses).

| Loss | IF | Method | Many | Medium | Few | All | Time |
|------|-----|--------|------|--------|------|------|------|
| CE | 10 | SGD | 75.6 | 62.8 | 48.2 | 60.8 | 696 (1.00×) |
| | | SAM | 76.7 | 64.4 | 49.0 | 61.9 | 1390 (2.00×) |
| | | ImbSAM | 74.0 | 61.4 | 54.6 | 62.4 | 2148 (3.09×) |
| | | **FaSO** | 77.1 | 65.4 | 49.7 | **62.6** | **898** (1.29×) |
| | 100 | SGD | 75.9 | 52.0 | 15.7 | 44.6 | 516 (1.00×) |
| | | SAM | 76.3 | 51.6 | 17.0 | 45.2 | 932 (1.81×) |
| | | ImbSAM | 76.1 | 49.1 | 20.0 | 45.6 | 1270 (2.47×) |
| | | **FaSO** | 77.2 | 53.9 | 16.2 | **45.8** | **670** (1.30×) |
| BCL | 10 | SGD | 71.7 | 64.5 | 59.5 | 64.7 | 1674 (1.00×) |
| | | SAM | 72.5 | 65.2 | 60.0 | 65.3 | 2896 (1.73×) |
| | | ImbSAM | 71.9 | 66.0 | 60.3 | 65.5 | 3482 (2.08×) |
| | | **FaSO** | 73.9 | 66.0 | 60.4 | **66.1** | **1804** (1.08×) |
| | 100 | SGD | 68.5 | 54.2 | 34.2 | 50.5 | 1098 (1.00×) |
| | | SAM | 68.1 | 53.5 | 37.1 | 51.3 | 1802 (1.64×) |
| | | ImbSAM | 68.0 | 52.9 | 40.0 | 52.2 | 2148 (1.96×) |
| | | **FaSO** | 71.1 | 57.5 | 36.3 | **53.1** | **1178** (1.07×) |

Table 13: Top-1 accuracy (%) (↑) results for *Many*, *Medium*, *Few*, and overall classes on CIFAR-100 LT with CE and CB losses, under an imbalance factor of 100. LookSAM-$k$ denotes the method where the SAM update is performed every $k$ steps. FaSO and Muon are highlighted in light gray for focused comparison.

| Loss | Method | Many | Medium | Few | All |
|------|--------|------|--------|------|------|
| CE | SAM | 77.5 | 51.1 | 15.8 | 44.9 |
| | **Muon** | 77.2 | 52.4 | 17.3 | 45.8 |
| | **FaSO** | 77.2 | 53.9 | 16.2 | **45.8** |
| | LookSAM-2 | 74.8 | 46.7 | 10.5 | 40.7 |
| | LookSAM-3 | 70.1 | 37.8 | 6.5 | 35.0 |
| | LookSAM-4 | 64.2 | 29.9 | 4.6 | 30.1 |
| CB | SAM | 75.4 | 50.6 | 19.0 | 45.4 |
| | **Muon** | 76.4 | 52.2 | 19.7 | **46.5** |
| | **FaSO** | 76.5 | 52.5 | 19.3 | 46.4 |
| | LookSAM-2 | 73.2 | 46.9 | 13.6 | 41.5 |
| | LookSAM-3 | 70.0 | 40.1 | 9.7 | 36.9 |
| | LookSAM-4 | 67.3 | 35.8 | 7.8 | 34.1 |

iNaturalist-2018 is a large-scale real-world long-tailed dataset that contains 437.5k training images from 8,142 species. Following mainstream protocols (Cui et al., 2019; Du et al., 2024), we adopt a ResNet-50 backbone trained for 90 epochs using CE loss. We compare SGD, SAM, Muon, and our proposed FaSO under the *Many*, *Medium*, and *Few* splits. As shown in Table 14, Muon and FaSO both outperform SGD and SAM across all class splits, with especially clear improvements on tail classes. These results demonstrate that curvature-aware optimization methods, such as Muon and FaSO, generalize effectively to more challenging large-scale long-tailed datasets.

Table 14: Top-1 accuracy (%) (↑) results for *Many*, *Medium*, *Few*, and overall classes on iNaturalist-2018 dataset. Muon and FaSO exhibit consistent improvements across all class splits. FaSO and Muon are highlighted in light gray for focused comparison.

| Method | Many | Medium | Few | All |
|--------|------|--------|-----|-----|
| SGD | 74.6 | 64.9 | 56.8 | 62.7 |
| SAM | 76.4 | 66.8 | 58.8 | 64.6 |
| **FaSO** | 77.5 | 67.9 | 59.8 | 65.7 |
| **Muon** | 78.0 | 68.4 | 60.6 | **66.3** |

### C.8 Deeper Understanding between Theoretical Analysis and FaSO Design

To demonstrate that the design of FaSO is grounded in the intrinsic training dynamics of long-tailed learning, we conducted an empirical analysis tracking the evolution of loss landscape geometry. Specifically, we monitored the Hessian trace of the least frequent class on CIFAR-100 LT under IF=100 across the training process for SGD, Muon, and FaSO. The results are presented in Table 15.

Table 15: Evolution of the Hessian trace on CIFAR-100 LT under imbalance factor of 100. Lower values indicate flatter minima, which correlate with better generalization.

| Epoch | SGD | Muon | FaSO |
|-------|-----|------|------|
| 80 | 664.6 | 636.1 | 691.5 |
| 120 | 956.1 | 450.8 | 615.2 |
| 180 | 1629.6 | 703.6 | 667.6 |
| 200 | 2137.5 | 536.0 | 514.1 |

These results show that, in the early training phase, both SGD and Muon exhibit relatively low traces. This suggests that during early exploration, the inherent stochasticity of SGD gradients provides sufficient noise to avoid sharp minima. Consequently, the complex orthogonalization operations of Muon incur computational overhead without offering significant geometric advantages during this period. This justifies FaSO's design choice to prioritize SGD in the early stages to maintain high computational efficiency while the optimization landscape is still being actively explored.

In later training stages, the trace for SGD increases dramatically, indicating convergence to a sharper landscape, which is known to harm tail-class generalization. In contrast, Muon maintains significantly lower trace values, supporting our analysis that Muon reduces the relative dominance of high positive-curvature modes and helps guide optimization toward flatter tail-class landscapes. This provides the motivation for FaSO to progressively increase its usage of the Muon optimizer as training advances. These observations confirm FaSO as a principled solution that combines SGD's early efficiency with Muon's capability to mitigate sharpness in the later stages.

### C.9 Further Analysis of Muon's Gradient-spectrum Reshaping

We further investigate how Muon reshapes the gradient spectrum and how this effect is related to tail-class optimization. For a gradient matrix $\mathbf{g}_t = \sum_{i=1}^{r_t} s_i u_i v_i^\top$, the Muon update can be written as $\mathbf{O}_t = \sum_{i=1}^{r_t} u_i v_i^\top$. Thus, Muon preserves the singular directions of the gradient while reducing the imbalance among singular components. Components with smaller singular values therefore receive more emphasis relative to already dominant components.

To make this explicit, consider two index sets of singular components: a dominant set $\mathcal{B}$, where $s_i \geq \beta$, and a suppressed set $\mathcal{A}$, where $s_i \leq \alpha$, with $0 < \alpha < \beta$. Since the singular update components $u_i v_i^\top$ are orthonormal under the Frobenius inner product, the relative strengthening ratios on $\mathcal{A}$ and $\mathcal{B}$ satisfy

$$R_\mathcal{A} = \frac{\|P_\mathcal{A} \mathbf{O}_t\|_F^2}{\|P_\mathcal{A} \mathbf{g}_t\|_F^2} = \frac{|\mathcal{A}|}{\sum_{i \in \mathcal{A}} s_i^2}, \qquad R_\mathcal{B} = \frac{\|P_\mathcal{B} \mathbf{O}_t\|_F^2}{\|P_\mathcal{B} \mathbf{g}_t\|_F^2} = \frac{|\mathcal{B}|}{\sum_{i \in \mathcal{B}} s_i^2}. \tag{16}$$

Comparing the two ratios gives

$$\frac{R_\mathcal{A}}{R_\mathcal{B}} = \frac{|\mathcal{A}|/\sum_{i \in \mathcal{A}} s_i^2}{|\mathcal{B}|/\sum_{i \in \mathcal{B}} s_i^2} = \frac{\frac{1}{|\mathcal{B}|}\sum_{i \in \mathcal{B}} s_i^2}{\frac{1}{|\mathcal{A}|}\sum_{i \in \mathcal{A}} s_i^2} \geq \frac{\beta^2}{\alpha^2} > 1. \tag{17}$$

This indicates that suppressed singular components are relatively strengthened more than dominant ones. In long-tailed learning, gradients can be dominated by head-class with high-frequency patterns, while tail-related signals may appear in less dominant spectral components. This shows that Muon prevents weaker components from being overwhelmed by the largest singular directions. Under the local view in Proposition 3.1, this effect is consistent with balancing updates across curvature directions and reducing the dominance of sharp directions. This helps explain Muon's association with flatter loss landscapes for tail classes.

Table 16: Spectral width ($\downarrow$) results for the rarest class on CIFAR-10 LT and CIFAR-100 LT with an imbalance factor of 100.

| Dataset | SGD | Muon |
|---|---|---|
| CIFAR-10 LT | 1284.81 | 226.35 |
| CIFAR-100 LT | 1845.68 | 710.42 |

To further examine the curvature structure of tail-class landscapes, we compute the Hessian spectral width $\lambda_{\max}(\mathbf{H}) - \lambda_{\min}(\mathbf{H})$ for the rarest class on both CIFAR-10 LT and CIFAR-100 LT under IF=100. A smaller spectral width indicates a more concentrated and less ill-conditioned local curvature profile. As shown in Table 16, Muon substantially reduces the spectral width for the rarest classes on both datasets. Together with the results in Table 3 and Fig. 3, this provides additional evidence that Muon is associated with smoother loss landscapes and better generalization for tail classes in long-tailed learning.

### C.10 Robustness to Training Duration

To further evaluate the robustness of the sinusoidal schedule, we conduct an additional ablation study by varying the total number of training epochs on CIFAR-100 LT with an imbalance factor of 100. Specifically, we consider a shortened training setting of 100 epochs and an extended training setting of 500 epochs under both CE and LA losses, and compare FaSO with SGD and Muon.

As shown in Table 17, FaSO consistently improves over SGD and remains competitive with Muon under both shortened and extended training settings. The improvements are observed across overall performance as well as medium- and few-class subsets, indicating that the effectiveness of the sinusoidal schedule does not diminish when the total training duration changes. These results demonstrate that the schedule is not overly sensitive to the absolute number of training epochs. By parameterizing the transition probability with the normalized progress, FaSO can adapt to different training budgets while still allocating orthogonalized updates at effective stages of optimization, thereby preserving its benefit for the generalization of tail classes.

Table 17: Top-1 accuracy (%) (↑) results on *Many, Medium, Few,* and overall classes on CIFAR-100 LT with an imbalance factor of 100, categorized by different training durations and loss functions (CE and LA). FaSO and Muon are highlighted in light gray for focused comparison.

| Epoch | Loss | Method | Many | Medium | Few | All |
|---|---|---|---|---|---|---|
| 100 | CE | SGD | 72.13 | 44.73 | 10.43 | 39.23 |
| | | **Muon** | 74.20 | 48.07 | 13.75 | 42.18 |
| | | **FaSO** | 75.37 | 48.93 | 13.00 | **42.49** |
| | LA | SGD | 64.20 | 50.60 | 30.20 | 46.52 |
| | | **Muon** | 65.77 | 51.83 | 32.50 | 48.28 |
| | | **FaSO** | 65.60 | 52.90 | 31.98 | **48.34** |
| 500 | CE | SGD | 76.67 | 51.80 | 17.58 | 45.57 |
| | | **Muon** | 78.63 | 54.33 | 19.50 | **47.69** |
| | | **FaSO** | 77.00 | 53.77 | 19.63 | 47.08 |
| | LA | SGD | 70.90 | 53.67 | 33.75 | 50.90 |
| | | **Muon** | 72.77 | 55.13 | 34.35 | 52.11 |
| | | **FaSO** | 69.10 | 57.37 | 36.33 | **52.47** |

## D    Discussions

**Additional related work.** Recent advancements in long-tailed recognition have diversified beyond traditional re-balancing techniques. In the realm of contrastive learning, GPaCo (Cui et al., 2024) identifies the bias of supervised contrastive loss towards high-frequency classes and introduces parametric learnable centers to rebalance optimization dynamics. For handling diverse test distributions, DirMixE (Yang et al., 2024) proposes a sophisticated mixture-of-experts strategy based on Dirichlet meta-distributions to capture both global and local label distribution variations. Furthermore, addressing the geometry of the loss landscape has become a pivotal direction; CC-SAM (Zhou et al., 2023b) argues that naive flattening is insufficient for long-tailed learning and proposes a class-conditional sharpness-aware minimization to robustify the classifier against parameter perturbations.

**Muon enhances representation learning.** Muon fundamentally improves representation learning by guiding optimization toward flatter minima. In long-tailed recognition, a critical representational failure mode is the tendency for minority classes to converge to sharp regions of the loss landscape, which undermines generalization capabilities. Muon addresses this challenge. By balancing update strength across positive-curvature modes via gradient orthogonalization, Muon reduces the dominance of sharp directions and guides optimization toward flatter solutions. Securing these flatter minima, as evidenced by improved loss-landscape metrics on tail classes, is essential for learning robust representations that generalize better to underrepresented data, going beyond mere improvements in convergence speed. Table 10 and Table 11 also provide more empirical evidence that Muon could improve the quality of learned representations.

**Regarding the CNC assumption.** The Correlated Negative Curvature (CNC) assumption provides a standard way to formalize that stochastic gradients contain informative components along directions of negative curvature (Daneshmand et al., 2018). It has been used in prior analyses of how stochastic gradients help optimization move away from sharp regions, including studies related to long-tailed learning (Rangwani et al., 2022; Zhou et al., 2023a). It has also been empirically examined across neural networks with varying widths and depths in practical settings (Zhu et al., 2019; Wang et al., 2020). However, modern neural-network landscapes are over-parameterized and structurally complex, with Hessian spectra exhibiting a bulk component together with isolated outlier directions (Ghorbani et al., 2019; Papyan, 2019). Moreover, sharpness-related geometry can vary with architecture, optimizer, batch size, and reparameterization (Dinh et al., 2017; Li et al., 2018). Therefore, CNC should be viewed as a local sufficient condition for a particular escape mechanism in this setting, rather than a complete global characterization of modern non-convex landscapes.

**Targeted design addressing long-tailed learning challenges.** Our work identifies and operationalizes a unique complementarity between gradient orthogonalization and long-tailed learning through two specialized contributions. First, our theoretical analysis demonstrates that Muon addresses optimization bottlenecks inherent to underrepresented data by balancing update strength across positive-curvature modes, reducing the relative dominance of sharp directions that hinder tail-class generalization. Second, building on this insight, we designed FaSO specifically for long-tailed training dynamics. FaSO progressively integrates Muon during critical later stages when SGD becomes more prone to stalling in sharp regions, thereby increasing tail-class generalization while mitigating computational overhead. This provides an efficient solution essential for large-scale imbalanced benchmarks.

**More insights on eigenvalues.** The minimum eigenvalue of the Hessian can also provide geometric information, as highly negative values typically indicate convergence to unstable saddle-like regions. To validate performance, we compute the minimum eigenvalues for the class with the fewest samples under Muon and SGD on both CIFAR-10 LT and CIFAR-100 LT under IF=100. On CIFAR-10 LT, the minimum eigenvalue under Muon is -110.33, which is substantially larger than the -316.30 observed under SGD. This trend is even more pronounced on the challenging CIFAR-100 LT dataset, where Muon achieved a $\lambda_{min}$ of -352.22 compared to -916.12 for SGD. This observation indicates significantly weaker negative curvature, corroborating Muon's effectiveness in escaping saddle-like regions for tail classes.

**FaSO as a trade-off between efficiency and generalization.** Our primary design objective for FaSO is to approximate Muon's generalization benefits while substantially mitigating its computational overhead. As shown in Table 1 and Table 2, FaSO exhibits a consistent performance pattern, reliably outperforming SGD and remaining competitive with Muon in terms of accuracy. Crucially, as shown in Table 4, it achieves these results while drastically reducing training costs compared to the significant overhead demands of Muon and SAM. Thus, FaSO successfully delivers its intended precise trade-off between high efficiency and robust generalization of tail classes.

