# OpenReview forum: "Fast Sharpness-escaping Optimization for Long-tailed Learning"
_TMLR — Under review for TMLR_

### Review · Reviewer_MhXu · 2026-04-26

**Summary Of Contributions:**

The paper proposes FaSO (Fast Sharpness-escaping Optimization), a novel optimization strategy designed for long-tailed learning. The authors first provide a theoretical analysis showing that the recently proposed Muon optimizer can help models escape sharp minima by amplifying updates along negative curvature directions. To address Muon's high computational overhead (caused by Newton-Schulz iterations), the authors design FaSO. FaSO dynamically schedules standard SGD and Muon updates using a sinusoidal probability function—relying predominantly on SGD early in training for efficient exploration, and increasingly on Muon later in training to escape sharp minima.

1. The theoretical motivation linking Muon's gradient orthogonalization to escaping sharp minima is sound and well-presented.
2. The empirical efficiency gains are excellent. FaSO dramatically reduces the computational overhead compared to SAM and standard Muon, while maintaining or even exceeding their generalization performance on tail classes.
3. Extensive experiments across multiple benchmarks (CIFAR-LT, ImageNet-LT, Places-LT) consistently demonstrate the effectiveness of the method.

**Audience:**

Yes

**Audience Explanation:**

The theoretical insights into the loss landscape geometry of the Muon optimizer, coupled with a highly practical and computationally efficient scheduling algorithm (FaSO) that directly benefits real-world long-tailed data training, will be relevant and interesting to this community.

**Claims And Evidence:**

Yes

**Claims Explanation:**

The theoretical claims are mathematically supported by Theorem 3.2 under the standard Correlated Negative Curvature (CNC) assumption. The empirical claims regarding generalization and computational efficiency are heavily supported by extensive experiments. The authors provided clear metrics (such as Hessian eigenvalue and trace analyses) to prove the models successfully found flatter minima, and measured runtime/FLOPs to prove the computational efficiency of FaSO against strong baselines like SAM and Muon.

**Requested Changes:**

Would strengthen the work:

1. Provide a brief discussion or ablation study on how the total number of training epochs ($T$) impacts the performance of the sinusoidal scheduling function. Does the schedule's effectiveness diminish if the training is significantly shortened or extended?

2.While the Correlated Negative Curvature (CNC) assumption is standard, please add a brief discussion regarding its potential limitations in highly non-convex modern neural networks to provide a more balanced theoretical view.

---

> ### Author Response · Authors · 2026-06-20
> **Response to Reviewer MhXu**
>
> > Provide a brief discussion or ablation study on how the total number of training epochs ($T$) impacts the performance of the sinusoidal scheduling function. Does the schedule's effectiveness diminish if the training is significantly shortened or extended?
>
> Thank you for this valuable suggestion. To better understand the robustness of the scheduling function, we conducted an additional ablation study on CIFAR-100 LT with IF=100 under both CE and LA losses. Specifically, we evaluated FaSO under a significantly shortened training setting of 100 epochs and an extended setting of 500 epochs. The results are summarized in the tables below:
>
> | Loss | Method | Many  | Med.  | Few   | All   |
> | ---- | ------ | :---: | :---: | :---: | :---: |
> | CE   | SGD    | 72.13 | 44.73 | 10.43 | 39.23 |
> | CE   | Muon   | 74.20 | 48.07 | 13.75 | 42.18 |
> | CE   | FaSO   | 75.37 | 48.93 | 13.00 | **42.49** |
> | LA   | SGD    | 64.20 | 50.60 | 30.20 | 46.52 |
> | LA   | Muon   | 65.77 | 51.83 | 32.50 | 48.28 |
> | LA   | FaSO   | 65.60 | 52.90 | 31.98 | **48.34** |
>
> *Tab.(a) Comparison on CIFAR-100 LT with 100 epochs.*
>
> | Loss | Method | Many  | Med.  | Few   | All   |
> | ---- | ------ | :---: | :---: | :---: | :---: |
> | CE   | SGD    | 76.67 | 51.80 | 17.58 | 45.57 |
> | CE   | Muon   | 78.63 | 54.33 | 19.50 | **47.69** |
> | CE   | FaSO   | 77.00 | 53.77 | 19.63 | 47.08 |
> | LA   | SGD    | 70.90 | 53.67 | 33.75 | 50.90 |
> | LA   | Muon   | 72.77 | 55.13 | 34.35 | 52.11 |
> | LA   | FaSO   | 69.10 | 57.37 | 36.33 | **52.47** |
>
> *Tab.(b) Comparison on CIFAR-100 LT with 500 epochs.*
>
> These results show that the sinusoidal schedule remains effective under both shortened and extended training budgets. FaSO consistently improves over SGD and achieves performance comparable to Muon, with clear gains on medium and tail classes. This suggests that the schedule is robust to changes in the total training duration.
>
> We have included this additional ablation study and discussion in Appendix C.10. Thank you again for this constructive comment.
>
> > While the Correlated Negative Curvature (CNC) assumption is standard, please add a brief discussion regarding its potential limitations in highly non-convex modern neural networks to provide a more balanced theoretical view.
>
> Thank you for this valuable suggestion. We have included the following discussion in Appendix D to clarify the limitations of the CNC assumption:
>
> Modern neural-network landscapes are over-parameterized and structurally complex, with Hessian spectra exhibiting a bulk component together with isolated outlier directions [a,b]. Moreover, sharpness-related geometry can vary with architecture, optimizer, batch size, and reparameterization [c,d]. Therefore, CNC should be viewed as a local sufficient condition for a particular escape mechanism in this setting, rather than a complete global characterization of modern non-convex landscapes.
>
> In the revised manuscript, the main theoretical interpretation is based on local curvature-mode balancing, while CNC is retained only as related background. Thank you again for this valuable suggestion.
>
> [a] An Investigation into Neural Net Optimization via Hessian Eigenvalue Density. ICML 2019.
>
> [b] Measurements of Three-Level Hierarchical Structure in the Outliers in the Spectrum of Deepnet Hessians. ICML 2019.
>
> [c] Visualizing the Loss Landscape of Neural Nets. NeurIPS 2018.
>
> [d] Sharp Minima Can Generalize For Deep Nets. ICML 2017.

---

### Review · Reviewer_WD2A · 2026-05-05

**Summary Of Contributions:**

The paper proposes FaSO, an optimization scheme that stochastically combines SGD and Muon: early iterations favor SGD, while Muon is used more frequently as training progresses. Experiments on long-tailed image datasets, including ImageNet and CIFAR, demonstrate that FaSO achieves performance comparable to Muon at a reduced computational cost.

**Strengths**

- The authors demonstrated empirically that Muon favors flatter minima relative to SGD.
- The empirical case for FaSO is convincing.

**Weakness**

The link between the theory and the empirical findings needs to be tightened.

The authors aimed to show theoretically that Muon optimization amplifies gradient components aligned with directions of negative curvature. Prior work has linked this property to faster escape from sharp local minima. However, as I understand it, the proof in Appendix B applies to any direction, and simply points to the fact that Muon maintains a gradient of larger magnitude towards the end of training. As such, the prior works cited do not seem the most relevant and might be misleading.

**Audience:**

Yes

**Audience Explanation:**

The algorithm seems easy to implement and provides a nice balance between the compute-efficiency of SGD and performance of Muon for long-tail tasks.

**Claims And Evidence:**

No

**Claims Explanation:**

Please see weakness: the claim of Muon amplifying gradients in negative eigen directions might be misleading.

**Requested Changes:**

Please revisit section 3.3 in response to the weakness.

---

> ### Author Response · Authors · 2026-06-20
> **Response to Reviewer WD2A**
>
> Thank you for the insightful comment. In the revised manuscript, we have refined our theoretical framing to focus on balancing updates across curvature directions and reducing the dominance of sharp directions. This refined perspective provides a tighter link to the empirical findings while maintaining the focus on landscape smoothness.
>
> To further address your concern, we added a discussion in Appendix C.9 on how Muon reshapes the singular structure of the gradient and how this effect is related to tail-class optimization. For a gradient matrix $\mathbf{g} _ t=\sum _ {i=1}^{r _ t}s _ i u _ i v _ i^\top$, the Muon update is $\mathbf{O} _ t=\sum _ {i=1}^{r _ t}u _ i v _ i^\top$. We then consider two index sets of singular components: a suppressed set $\mathcal{A}$, where $s _ i\le \alpha$, and a dominant set $\mathcal{B}$, where $s _ i\ge \beta$, with $0<\alpha<\beta$. Since the singular update components $u _ i v _ i^\top$ are orthonormal under the Frobenius inner product, the relative strengthening ratios on $\mathcal{A}$ and $\mathcal{B}$ satisfy
> $$
> R _ {\mathcal{A}}=\frac{|P _ {\mathcal{A}}\mathbf{O} _ t| _ F^2}{|P _ {\mathcal{A}}\mathbf{g} _ t| _ F^2}=\frac{|\mathcal{A}|}{\sum _ {i\in\mathcal{A}}s _ i^2},
> \qquad
> R _ {\mathcal{B}}=\frac{|P _ {\mathcal{B}}\mathbf{O} _ t| _ F^2}{|P _ {\mathcal{B}}\mathbf{g} _ t| _ F^2}=
> \frac{|\mathcal{B}|}{\sum _ {i\in\mathcal{B}}s _ i^2}.
> $$
>
> Comparing these ratios gives
> $$
> \frac{R _ {\mathcal{A}}}{R _ {\mathcal{B}}}=\frac{\frac{1}{|\mathcal{B}|}\sum _ {i\in\mathcal{B}}s _ i^2}{\frac{1}{|\mathcal{A}|}\sum _ {i\in\mathcal{A}}s _ i^2}
> \ge\frac{\beta^2}{\alpha^2}> 1.
> $$
>
> This indicates that suppressed singular components are relatively strengthened more than dominant ones. In long-tailed learning, gradients can be dominated by head classes and high-frequency patterns, while tail-related signals may appear in less dominant spectral components. This shows that Muon helps prevent weaker components from being overwhelmed by the largest singular directions, which is consistent with balancing curvature updates and mitigating sharp directions.
>
> We also added the Hessian spectral width $\lambda_{\max}(\mathbf{H})-\lambda_{\min}(\mathbf{H})$ for the rarest class on CIFAR-10 LT and CIFAR-100 LT under IF=100. A smaller spectral width indicates a more concentrated and less ill-conditioned local curvature profile. As shown below, Muon effectively reduces the spectral width compared with SGD.
>
> | Dataset      |     SGD |   Muon |
> | ------------ | ------: | -----: |
> | CIFAR-10 LT  | 1284.81 | 226.35 |
> | CIFAR-100 LT | 1845.68 | 710.42 |
>
> These analyses provide additional evidence that Muon is associated with smoother tail-class landscapes and better generalization in long-tailed learning. We have revised the relevant discussion and Appendix C.9 accordingly. We sincerely appreciate your thoughtful suggestion.

---

### Review · Reviewer_4XYK · 2026-06-08

**Summary Of Contributions:**

This paper studies the Muon optimizer in the long-tailed learning from a geometric perspective. The authors argue that its polar factor update can help escape from sharp loss-landscape regions associated with tail classes. The authors further analyze the computational overhead of Muon and propose FaSO, which randomly alternates between SGD and Muon while increasing the probability of Muon updates later in training. Experiments across multiple image long-tailed classification benchmarks show improved accuracy over SGD and reduced computation cost compared to the original Muon optimizer.

**Strengths**:
- The paper addresses an interesting problem about Muon optimizer.
- The proposed FaSO is simple and intuitive with clear empirical benefit over several benchmarks.

**Weaknesses**  (See details in Evidence & Significance section):
- The central theoretical claim is invalid as stated.
- The experiments do not directly verify the proposed negative-curvature escape mechanism. Some performance claims are stronger than the reported results, and several numerical and reporting inconsistencies require correction.

**Audience:**

Yes

**Audience Explanation:**

Yes, if the claims are supported by solid evidence, it would help advance the understanding of Muon optimizer in the long-tailed learning.

**Claims And Evidence:**

No

**Claims Explanation:**

- The main theoretical claim, Theorem 3.2, is flawed under the stated assumptions.
	1. Eq. (26) of the proof incorrectly states that $\sum_{i,j}(1-s_is_j)m_im_j =m^\top(I-ss^\top)m$. The correct matrix expression is $m^\top(\mathbf{1}\mathbf{1}^\top-ss^\top)m$, where $\mathbf{1}$ is the all-ones vector. Although $I-ss^\top$ is positive semidefinite when $\lVert s\rVert_2=1$, the matrix $\mathbf{1}\mathbf{1}^\top-ss^\top$ is generally indefinite. Lemma B.2 therefore does not apply, and Eq. (27)-(28) do not follow.
		2. As a counterexample, we can construct $G=\begin{pmatrix}\sqrt{3}/2 & 0 \\\ 0 & 1/2\end{pmatrix}, V=\frac{1}{\sqrt{2}}\begin{pmatrix}1 & 0 \\\ 0 & -1\end{pmatrix}$. By Eq. (2), the Muon update $O=I$ and $\operatorname{proj}\_{\mathrm{Muon}}=\langle V,I\rangle\_F=0$. In contrast, $\operatorname{proj}\_{\mathrm{SGD}}=\langle V,G\rangle_F=\frac{\sqrt{3}-1}{2\sqrt{2}}\approx 0.259$ and hence $\operatorname{proj}_{\mathrm{SGD}}^2\approx 0.067$. A deterministic stochastic-gradient distribution concentrated at this $G$ satisfies the pointwise CNC lower bound for any $\gamma\leq 0.067$, while the Muon projection is zero. Thus, CNC alone does not imply the first inequality in Theorem 3.2.
- Sec. 3.4 FLOPs calculation in Eq. (6) is inconsistent: the text states that a standard linear-layer computation costs approximately $6mnL$ FLOPs, but Equations (6)-(7) divide the cost by $2mnL$.
- The empirical sharpness discussion uses $\lambda_{\max}$ and $\operatorname{Tr}(H)$, which characterize large positive curvature. The theoretical argument instead concerns the eigenvector associated with $\lambda_{\min}$. But the connection between increasing the projection along a most-negative-curvature direction and reducing $\lambda_{\max}$ or $\operatorname{Tr}(H)$ is not established.
- Empirical claims are only partially supported.
	- Muon usually improves overall accuracy over SGD, and FaSO often retains much of Muon's performance at lower runtime.
	- However, the stronger universal claims are not discharged. For example, the statement that Muon has an edge over SAM "across all conditions" is false. For example, on CIFAR-100 LT, IF=10 with CE, Muon obtains 61.8 overall versus SAM's 61.9. There are many more such examples in Table 1.
- Various inconsistencies in reporting
	- Table 12's LA rows duplicate Table 1's CB rows.
	- The same CE/SAM setting differs across Tables 1, 11, and 12.
	- The learning-rate description is contradictory: an initial rate of $3\times10^{-4}$ is said to be "decayed to 0.01."

**Requested Changes:**

Critical:
- Fix or replace Theorem 3.2.
- Provide a valid theoretical bridge if the paper retains claims about escaping sharp minima or reaching flatter solutions.
- Correct the FLOP analysis using a consistent baseline and double check the numerical ResNet example.
- Qualify the universal performance claims (Muon wins in all conditions). Add scope sentences to bound the claim properly.

Minor:
- Correct the inconsistent tables and learning-rate description.

---

> ### Author Response · Authors · 2026-06-20
> **Response to Reviewer 4XYK (1/2)**
>
> > The main theoretical claim, Theorem 3.2, is flawed under the stated assumptions. Fix or replace Theorem 3.2.
> > The empirical sharpness discussion uses $\lambda_{\max}$ and $\operatorname{Tr}(H)$, which characterize large positive curvature. The theoretical argument instead concerns the eigenvector associated with $\lambda_{\min}$. But the connection between increasing the projection along a most-negative-curvature direction and reducing $\lambda_{\max}$ or $\operatorname{Tr}(H)$ is not established. Provide a valid theoretical bridge if the paper retains claims about escaping sharp minima or reaching flatter solutions.
>
> Thank you for the careful and technically precise comment. We acknowledge that the quadratic-form simplification used in the original proof does not hold in general, and therefore the resulting nonnegativity claim is not guaranteed under the stated assumptions. Accordingly, we have replaced the original Theorem 3.2 and updated the corresponding claim.
>
> To address your concern, we have replaced the original theorem with a local positive-curvature mode-balancing proposition. This proposition does not rely on $\lambda_{\min}$ or the CNC assumption. Instead, it analyzes how the Muon polar update balances deterministic update strength across positive-curvature modes, which is more aligned with the empirical sharpness metrics $\lambda_{\max}$ and $\mathrm{Tr}(\mathbf{H})$. In the local curvature view, each mode corresponds to a Hessian eigen-direction, and $h_i$ denotes the positive curvature value of the $i$-th mode. Let $\alpha_i^{\mathrm{SGD}}$ and $\alpha_i^{\mathrm{Muon}}$ denote the deterministic update strengths assigned to this mode by SGD and Muon, respectively. The revised proposition shows that, for two modes $a$ and $b$ with $h_a>h_b>0$,
> $$\frac{\alpha_a^{\mathrm{Muon}}}{\alpha_b^{\mathrm{Muon}}}=
> \sqrt{\frac{h_a}{h_b}}<\frac{h_a}{h_b}=
> \frac{\alpha_a^{\mathrm{SGD}}}{\alpha_b^{\mathrm{SGD}}}.$$
>
> Thus, compared with SGD, Muon compresses the gap between sharp and relatively flat directions, making the update less dominated by the sharpest positive-curvature modes. This provides a more direct bridge to the empirical sharpness discussion. The empirical metrics $\lambda_{\max}$ and $\mathrm{Tr}(\mathbf{H})$ characterize complementary aspects of Hessian sharpness: $\lambda_{\max}$ captures the largest positive-curvature direction, while $\mathrm{Tr}(\mathbf{H})$ reflects the overall curvature scale measured in our experiments. Therefore, this is naturally aligned with the observed reductions in Hessian sharpness for tail classes.
>
> We have revised the manuscript accordingly by replacing the theorem, updating Appendix B, and rephrasing the theoretical interpretation throughout the paper as positive-curvature mode balancing. This refinement provides a more precise theoretical basis while maintaining the focus on landscape smoothness and empirical sharpness reduction. Thank you again for pointing out this important issue.
>
> > Sec. 3.4 FLOPs calculation in Eq. (6) is inconsistent: the text states that a standard linear-layer computation costs approximately 6mnL FLOPs, but Equations (6)-(7) divide the cost by 2mnL. Correct the FLOP analysis using a consistent baseline and double check the numerical ResNet example.
>
> Thank you for your helpful comment. We have revised the FLOP analysis in Sec. 3.4. Specifically, $2mnL$ corresponds to the forward-only cost of a linear layer, while our overhead analysis is based on the full training cost, including the forward pass and the two dominant backward matrix multiplications. This gives approximately $2mnL+2mnL+2mnL=6mnL$. Therefore, the corresponding overhead formulas should use $6mnL$ as the denominator.
>
> The corrected linear-layer overhead becomes
> $$\Delta F_{\text{linear}}=\frac{T\cdot 6mn^2}{L\cdot 6mn}=\frac{Tn}{B}\in O\left(\frac{Tn}{B}\right).$$ Similarly, the convolutional-layer overhead becomes
> $$\Delta F_{\text{conv}}=\frac{T\cdot 6mn^2}{L\cdot 6mn}=\frac{T C_{\text{in}}k^2}{B H_{\text{out}}W_{\text{out}}}\in O\left(\frac{T C_{\text{in}}k^2}{B H_{\text{out}}W_{\text{out}}}\right).$$
>
> We have also checked the numerical ResNet example. The reported additional overhead of 183% is computed using the correct formula. Thank you again for pointing this out.

---

> ### Author Response · Authors · 2026-06-20
> **Response to Reviewer 4XYK (2/2)**
>
> > Empirical claims are only partially supported. Qualify the universal performance claims (Muon wins in all conditions). Add scope sentences to bound the claim properly.
>
> Thank you for this insightful comment. We have revised the manuscript to better qualify our empirical claims and avoid overly universal statements. The revised text now describes Muon as generally improving over SGD and achieving competitive or better performance than SAM across the evaluated long-tailed settings, with the clearest gains on tail classes and under severe imbalance. We also clarify that FaSO achieves a favorable accuracy-efficiency trade-off by generally matching Muon’s performance while substantially reducing the runtime overhead of full Muon updates.
>
> In addition, we added scope sentences to emphasize that the results indicate overall trends rather than uniform dominance in every setting. These revisions better align the empirical discussion with the reported results. Thanks again for the helpful suggestion.
>
> > Various inconsistencies in reporting. Correct the inconsistent tables and learning-rate description.
>
> Thank you for pointing out these details. We have carefully checked the reported results and corrected the related entries in Tables 11 and 12. We have also corrected the learning-rate description in the implementation details: the experiments use an initial learning rate of 0.1. Thank you again for the careful check.